# Repair of multiple simultaneous double-strand breaks causes bursts of genome-wide clustered hypermutation

**Cynthia J. Sakofsky**[1], **Natalie Saini**[1], **Leszek J. Klimczak**[2], **Kin Chan**[1¤], **Ewa P. Malc**[3], **Piotr A. Mieczkowski**[3], **Adam B. Burkholder**[2], **David Fargo**[2], **Dmitry A. Gordenin**[1]*

**1** Genome Integrity and Structural Biology Laboratory, National Institute of Environmental Health Sciences, US National Institutes of Health, North Carolina, United States of America, **2** Integrative Bioinformatics Support Group, National Institute of Environmental Health Sciences, US National Institutes of Health, North Carolina, United States of America, **3** Department of Genetics, Lineberger Comprehensive Cancer Center, University of North Carolina, Chapel Hill, North Carolina, United States of America

¤ Current address: Department of Biochemistry, Microbiology and Immunology, University of Ottawa, Ottawa, Ontario, Canada
* gordenin@niehs.nih.gov

**Data Availability Statement:** All relevant data are within the paper and its Supporting Information files. Whole genome sequence of ySR128 were deposited to GenBank (https://www.ncbi.nlm.nih.

## Abstract

A single cancer genome can harbor thousands of clustered mutations. Mutation signature analyses have revealed that the origin of clusters are lesions in long tracts of single-stranded (ss) DNA damaged by apolipoprotein B mRNA editing enzyme, catalytic polypeptide-like (APOBEC) cytidine deaminases, raising questions about molecular mechanisms that generate long ssDNA vulnerable to hypermutation. Here, we show that ssDNA intermediates formed during the repair of gamma-induced bursts of double-strand breaks (DSBs) in the presence of APOBEC3A in yeast lead to multiple APOBEC-induced clusters similar to cancer. We identified three independent pathways enabling cluster formation associated with repairing bursts of DSBs: 5′ to 3′ bidirectional resection, unidirectional resection, and break-induced replication (BIR). Analysis of millions of mutations in APOBEC-hypermutated cancer genomes revealed that cancer tolerance to formation of hypermutable ssDNA is similar to yeast and that the predominant pattern of clustered mutagenesis is the same as in resection-defective yeast, suggesting that cluster formation in cancers is driven by a BIR-like mechanism. The phenomenon of genome-wide burst of clustered mutagenesis revealed by our study can play an important role in generating somatic hypermutation in cancers as well as in noncancerous cells.

## Introduction

Cellular DNA mostly exists in double-stranded (ds) form, which prevents irreversible breakage caused by chemical or enzymatic lesions and provides the template for error-free repair when one of the two DNA strands is damaged or broken [1]. However, major cellular processes such as DNA replication, repair, and transcription require single-stranded (ss) DNA intermediates,

gov/nuccore/), accession numbers CP036470-CP036486. Yeast sequencing data were deposited to SRA (https://www.ncbi.nlm.nih.gov/sra/PRJNA528224).

**Funding:** This work was supported by the US National Institute of Health Intramural Research Program Project Z1AES103266 to DAG. The funders had no role in study design, data collection and analysis, decision to publish, or preparation of the manuscript.

**Competing interests:** The authors have declared that no competing interests exist.

**Abbreviations:** AdenoCA, adenocarcinoma; AP site, apurinic/apyrimidinic site; APOBEC, apolipoprotein B mRNA editing enzyme, catalytic polypeptide-like; BIR, break-induced replication; BSD, blasticidin; Can-Low-Ade, synthetic medium with low adenine and lacking arginine but supplemented with 60 mg/L canavanine; Cen II, centromere of the chromosome II; CG, C- and/or G-containing; CNV, copy number variation; Com-Low-Ade, complete synthetic medium with 15 μg/ml concentration of adenine; dbGaP, Database of Genotypes and Phenotypes; DDR, DNA damage response; ds, double-stranded; DSB, double-strand break; HR, homologous recombination; LOH, loss of heterozygosity; PCAWG, Pan Cancer Analysis of Whole Genomes project; PFGE, pulse-field gel electrophoresis; SCC, squamous cell carcinoma; SNV, single-nucleotide base substitution variant; ss, single-stranded; TCC, transitional cell carcinoma; Tel2, telomere of the chromosome 2; WT, wild type; YPG, Yeast extract Peptone Glycerol.

but these are usually kept to the minimum levels necessary for a transaction to function. Special systems can sense abnormally long and/or persistent ssDNA formed during replication or DNA repair and trigger a DNA damage response (DDR), activating repair or cell-death pathways [2]. If, however, excessive ssDNA escapes surveillance, it can compromise genome integrity, resulting in rearrangements and/or in hypermutation [3]. Localized hypermutation can occur when transient stretches of ssDNA with multiple lesions are copied by translesion synthesis [4–6]. The ssDNA mutation rates can exceed by up to a thousand-fold the rates in the rest (ds part) of the genome, thus forming tightly spaced clusters of mutations, also termed "kataegis." The most striking form of clustered mutagenesis has been associated with ssDNA-specific apolipoprotein B mRNA editing enzyme, catalytic polypeptide-like (APOBEC) cytidine deaminases normally acting as a part of the innate immunity by converting cytosines to uracils in ssDNA intermediates of retroviruses and retrotransposons [7, 8] that accidentally gain access to genomic DNA in human cancers [5, 9–12]. APOBEC-mutated cytosines in clusters are often found in the characteristic trinucleotide context -tCw (mutated nucleotide capitalized; w is either thymine or adenine). Moreover, within a cluster, there is at least one stretch of several mutated cytosines belonging to the same DNA strand (strand-coordinated stretch), and often the entire cluster is strand-coordinated (Fig 1).

Strand-coordinated clusters can span over 100 kb in a single cancer genome. Moreover, cancers can contain hundreds of APOBEC-induced mutation clusters with varying extents of strand coordination. Altogether, this indicates the existence of multiple, abnormally long persistent stretches of ssDNA prone to hypermutation occurring throughout a cancer genome lineage; however, the sources of such ssDNA in cancers remain to be elucidated.

Studies in yeast models have indicated that long clusters of multiple mutations can stem from ssDNA intermediates formed during repair of a double-strand break (DSB) by homologous recombination (HR) (reviewed in [4] and Fig 1). Specifically, ssDNA can arise from unidirectional resection and/or break-induced replication (BIR) (Fig 1A) or from bidirectional resection (Fig 1B).

Clusters can be formed in ssDNA generated by 5′ to 3′ bidirectional DNA resection, creating long ssDNA stretches that act as a substrate for hypermutation by an ssDNA-specific mutagen. In support of such a mechanism, bases damaged by a mutagen around a site-specific DSB were found to be mutated in the top strand on one side of a break and in the bottom strand on the opposite side of the break ([6, 13] and Fig 1B). Clusters with switching strand coordination conforming to a bidirectional resection mechanism were also observed in yeast grown in the presence of an alkylating agent capable of inducing base damage in ssDNA, as well as infrequent DSBs that may lead to formation of ssDNA [5]. Another HR repair pathway capable of generating clustered mutations in the presence of an ssDNA-specific mutagen is BIR. Strand-coordinated clusters of damage-induced mutations were found along the tracks of BIR synthesis near a site-specific DSB, where ssDNA was expected to form because of unusual replication that accumulated a stretch of long ssDNA behind its replication bubble. BIR clusters also spanned the region flanking the DSB cut site where DNA was resected 5′ to 3′ in the preceding steps of BIR synthesis ([14] and Fig 1A). Importantly, yeast clusters associated with ssDNA formed by transcription [15] or at replication forks [5, 16] were much smaller than clusters associated with DSBs.

Colocalization of APOBEC mutation clusters with breakpoints of chromosomal rearrangements in human cancers has been documented in several studies [5, 12, 17]. Based on this and on the fact that clusters can span across many kb, DSBs could be a major source of ssDNA prone to hypermutation in cancers. In addition, bursts of DSBs occurring in the same or in several subsequent cell generations were implicated in chromothripsis, a common feature of cancers characterized by catastrophic multiple rearrangements in the cancer genome [18].

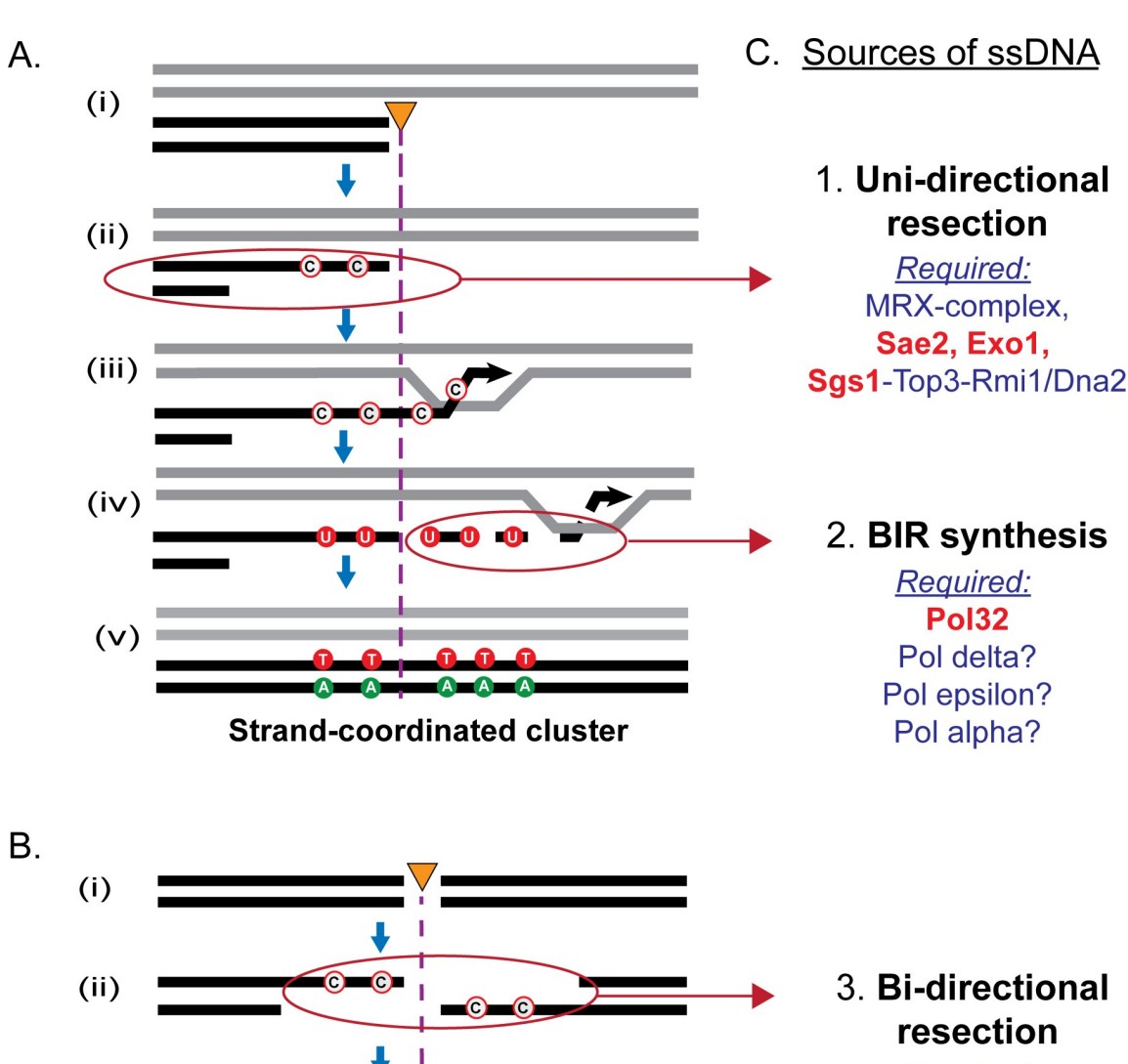

**Fig 1. DSB repair pathways generating ssDNA vulnerable to hypermutation by APOBEC cytosine deaminase.** The figure includes the following symbols designating nucleotides, mutations, and DNA strands: "unfilled red circles," nonmutated cytosines; "green-filled circles," contains a base replacing mutated guanines; "red-filled circles," contains a base replacing mutated cytosines; "black arrowheads," 3′ end of DNA strand extended by DNA polymerase; "orange triangle with purple dotted line," position of DSB; "C," cytosine; "A," adenine; "U," uracil; "T," thymine. On both panels, C to U changes are due to APOBEC cytosine deamination in the absence of uracil DNA glycosylase. (A) ssDNA formed during BIR repairing a one-ended DNA break. (i) Black lines show one-ended DNA break, and gray lines show a homologous template for repair. (ii) ssDNA generated from 5′ to 3′ end resection, resulting in a 3′ overhang. (iii) Invasion into a homologous template that initiates DNA synthesis. (iv) Progression of DNA synthesis generating a long ssDNA tail behind the migrating BIR replication bubble. Cytosines in ssDNA convert to uracil after deamination by APOBEC. (v) Completed repair of the DNA break by asynchronous lagging strand synthesis restoring DNA to ds form. (Not illustrated is a resolution of joint molecules.) C-coordinated clustered mutations result following one round of cellular replication (not illustrated) in which adenines were inserted opposite uracils, resulting in C:G to T:A mutations. If BIR initiated repair from the opposite side of a DNA break, G-coordinated clusters in the top strand would result from cytosine deamination in ssDNA generated on the bottom DNA strand (not illustrated). (B) ssDNA formed from bidirectional resection at a DSB. (i) DSB. (ii) 5′ to 3′ end resection generating ssDNA on both sides of the DNA break. (iii) Cytosines in ssDNA on the top (left) and bottom (right) DNA strands are converted to uracil by APOBEC

deamination. (iv) dsDNA is restored using a homologous template (not illustrated), followed by a round of replication generating a 5′ C single-switch cluster. (C) Three sources of ssDNA associated with HR repair promoting clustered mutations: unidirectional resection, BIR synthesis, and bidirectional resection. The main proteins required to generate each source of ssDNA is listed. Red font indicates diagnostic genetic defects used in this study. APOBEC, apolipoprotein B mRNA editing enzyme, catalytic polypeptide-like; BIR, break-induced replication; ds, double-stranded; DSB, double-strand break; HR, homologous recombination; ss, single-stranded.

Therefore, we propose that the large number of clusters in individual cancer genomes could also be caused by damage in extensive ssDNA associated with bursts of DSBs. To evaluate this hypothesis, we generated multiple simultaneous DSBs scattered throughout the yeast genome in the presence of the ssDNA-specific human APOBEC3A cytidine deaminase. It was established [19] that the A3A-like mutation signature prevails across the genomes and in the mutation clusters of APOBEC-hypermutated tumor types that were used in this study for comparison with yeast data (see below). Also, A3A-like signatures prevailed in bursts of APOBEC mutagenesis during propagation of cancer cell lines as revealed by [20]. This setup allowed us to test whether the genome of a eukaryotic yeast cell can tolerate multiple simultaneous stretches of ssDNA, which are sufficiently persistent to allow an ssDNA-specific mutagen to generate a large number of clustered mutations. We show here that bursts of DSBs can result in an abundance of clustered mutations arising in a single cell throughout its genome. We also revealed three independent sources of ssDNA that can simultaneously operate in a single cell to promote multiple clusters: (i) bidirectional DNA resection, (ii) long unidirectional resection, and (iii) BIR synthesis. Using the strand-assignment patterns of clustered mutations arising from these ssDNA sources in yeast, we then scrutinized over 10,000 mutation clusters in cancers with a high level of APOBEC mutagenesis and found that clusters in cancers matched patterns of clustered mutations in resection-defective yeast, suggesting that bidirectional resection does not play a major role in cluster formation, but rather, clusters in cancer are formed by a BIR-like mechanism.

## Results

### Bursts of DSBs after gamma irradiation in yeast expressing A3A lead to high levels of mutagenesis and multiple nonselected clusters

We tested whether bursts of DSBs could result in the vast formation of hypermutable ssDNA that leads to clustered mutations using *Saccharomyces cerevisiae* yeast strains harboring a triple reporter system (*CAN1-ADE2-URA3*) located in the midregion of chromosome II (Fig 2A).

This yeast expressed APOBEC3A (A3A) under the regulation of a doxycycline-inducible TET-promoter on a centromeric vector [16]. While a similar construct expressing APOBEC3B was also highly mutagenic in yeast, we chose A3A for these experiments because, based on our prior experience [19], A3A caused lower level of background mutagenesis in proliferating yeast but produced similar size and density of mutation clusters in ssDNA artificially generated at uncapped telomeres. The strains also lacked Ung1; therefore, uracils in ssDNA were not converted to apurinic/apyrimidinic sites (AP sites) [21]. All uracils formed by A3A-induced deamination of cytosines would result in C:G to T:A transitions in a tCw context with even greater preference for the A3A-specific trinucleotide context ytCa (y = T or C; mutated base capitalized) [19], leaving a permanent mark on transient regions of persistent ssDNA. Bursts of DSBs were formed from exposure of diploid and haploid yeast cultures arrested at the G2 phase of the cell cycle to either 80 krad or 40 krad, respectively. This resulted in approximately 100 DSBs per genome in diploids, with about 40% survival, and 50 DSBs per genome in haploids, with about 20% survival, consistent with previous studies [22, 23] (S1A and S1B Table). Since the strains with triple reporter (*CAN1-ADE2-URA3*) were canavanine-sensitive and Ade

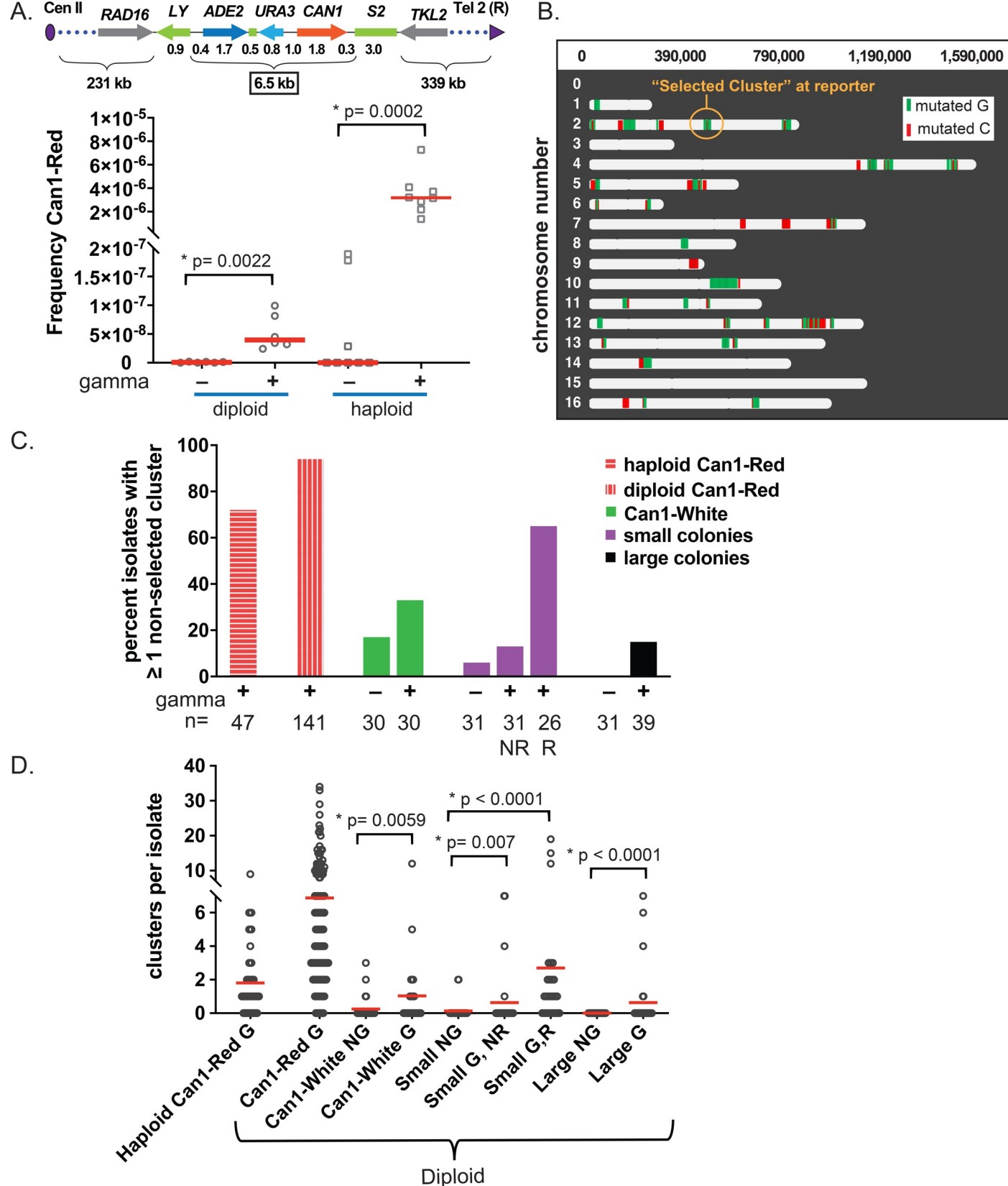

**Fig 2. Multiple nonselected clusters with APOBEC signature mutations in gamma-exposed WT yeast.** (A) Top—schematic of reporter system used to select Can1-Red mutants. Cassette was inserted into *LYS2* and has a total length of 6.5 kb. Distances of ORFs and intergenic regions from LY to S2 are shown below schematic and are in units of kb. The number of essential genes from Cen II to LY is 21 and from S2 to Tel2 is 35 (not illustrated). Bottom—graph showing Can1-Red mutant frequencies in gamma-exposed and in nonexposed diploid and haploid yeast. Can1-Red mutants were selected because they indicate coincident inactivation of closely spaced genes (*CAN1* and *ADE2*) and therefore were expected to have at least one mutation cluster. Horizontal bars show median mutation frequencies. Gray circles and gray squares indicate values from independent experiments using diploid and haploid yeast, respectively. *P*-values from two-tailed Mann–Whitney test comparing mutation frequencies in gamma versus no gamma are shown. Source data in S1C Table. (B) Mutations in the genome of an example diploid yeast Can1-Red mutant (7_24_CA_2) obtained after 80 krad of gamma irradiation. The isolate harbors 35 clusters with a total of 1,247 mutations ranging in length from 5,700 bp to 113,960 bp. Red and green bars indicate clusters with mutated C's and G's, respectively. The cluster spanning the reporter region, called "Selected Cluster," is circled in yellow. (C) Percent of isolates with at least one nonselected cluster. A nonselected cluster is a cluster that does not overlap the triple reporter region in chromosome II. The number of isolates from each yeast colony type are shown below the graph. "NR" = no rearranged chromosomes and "R" = rearranged chromosomes as determined by PFGE. Source data in S2C Table. (D) Numbers of nonselected clusters per isolate. Red line indicates the mean value. "G," gamma exposure; "NG," no gamma exposure; "R," rearranged chromosomes; "NR," no rearranged chromosomes. Labels on x-axis indicate yeast colony type. *P*-values from Fisher's Exact test comparing clusters per isolate in gamma- versus non-gamma–exposed cells are shown. Source data in S2C Table. (See also S1, S2 and S3 Tables and S2 Fig). APOBEC, apolipoprotein B mRNA editing enzyme, catalytic polypeptide-like; Cen II, centromere of the chromosome II; PFGE, pulse-field gel electrophoresis; Tel2, telomere of the chromosome 2; WT, wild type.

+ (capable of growing without adenine and white in color), it allowed convenient selection of simultaneous inactivation of *CAN1* and *ADE2*. Colonies of Can^R *ade2* double-inactivation mutants were identified on the medium containing canavanine by red color (pigmentation resulting from complete or partial *ADE2* inactivation) and will be referred to as Can1-Red hereafter. Importantly, in diploids and haploids, these levels of gamma exposure resulted in about a 70-fold increase in the frequency of coincident inactivation of the closely spaced *CAN1* and *ADE2* genes (Fig 2A, S1C Table). We used diploid strains to allow survival of isolates with the highest mutation loads, which would include recessive lethal mutations. In haploids, Can1-Red would result from a pair of mutations, while in diploids, it would require a loss of heterozygosity (LOH) following mutagenesis. Since gamma irradiation has been shown to induce LOH at high frequencies, we expected a detectable fraction of a double inactivation of the *CAN1* and *ADE2* in diploids. Since these colonies likely had mutations in the closely spaced *CAN1* and *ADE2* genes, they were likely to have at least one mutation cluster per genome, which was confirmed by whole-genome sequencing (S2C Table). Surprisingly, in the majority of isolates, we found up to 34 additional clusters in other regions of the yeast genome (called "nonselected" clusters), in addition to the clustered mutation that spanned the triple reporter region (called "selected clusters"). In Can1-Red diploids and haploids, the percent of isolates that had at least one nonselected cluster was over 70% (Fig 2B, 2C and 2D). These nonselected clusters ranged from 20 to over 223,000 bps, with a median value of 23,842, and had 2 to 854 mutations per cluster with a median value of 16 in diploids. In haploids, the lengths of nonselected clusters were between 25 to 32,064 bps with a median value of 8,682 and with 2 to 37 mutations per cluster with a median value of 6 (S2A, S2B and S2C Table and S3 Table, S1 Data). We found that nonselected clusters in Can1-Red diploid yeast were not localized to any particular genomic region, but rather, clusters were spread across the chromosomes in proportion to chromosome size (S6 Fig, S4F Table).

Importantly, clusters in all Can1-Red isolates from gamma-irradiated yeast were 4- to 5-fold–enriched with an A3A-like mutation signature, indicating that clusters represented genomic regions of transient hypermutable ssDNA (S2A and S2B Table). Taking into consideration that not all ssDNA regions could have been mutated by A3A, clusters would allow only a minimum estimate of the total length and percentage of hypermutable ssDNA formed in yeasts repairing bursts of DSBs. About 0.9% of the genome persisted as hypermutable DNA in diploids with a median total length of 210 kb and 0.2% of the genome in haploids, with a median total length of 28 kb (Fig 3, S4A and S4B Table).

We noticed that there was a strong correlation between the number of clustered mutations, total length of clusters, and density of clustered mutations of individual sequenced isolates,

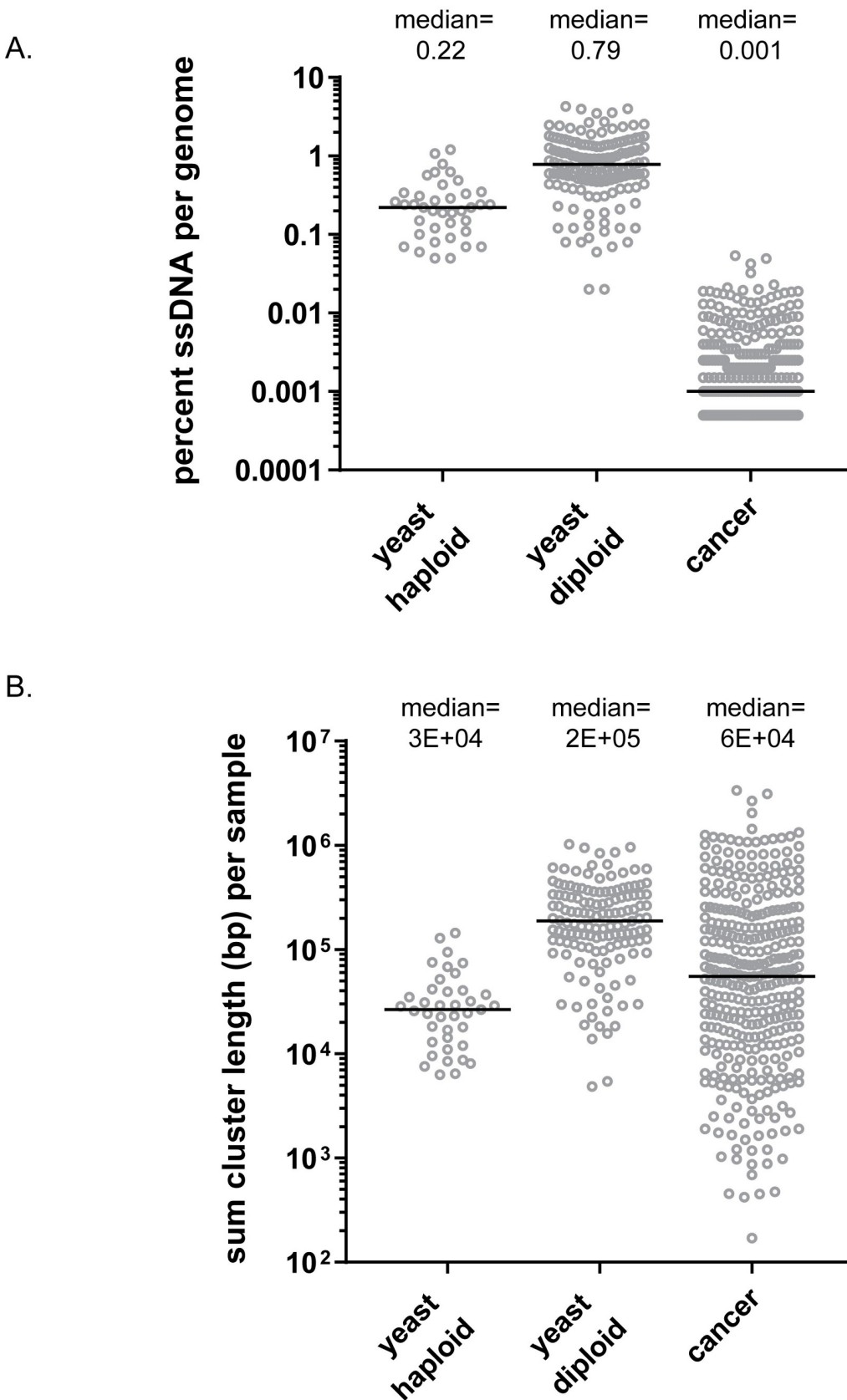

**Fig 3. Estimates of the hypermutable ssDNA per genome.** (A) The percentages of ssDNA per genome in WT yeast haploid and diploid Can1-Red colonies and APOBEC-enriched human cancer types, including Bladder-TCC, Breast-

AdenoCA, Cervix-SCC, Head-SCC, Lung-AdenoCA, and Lung-SCC. Percent ssDNA is calculated based on the following equation: total sum of cluster lengths per genome/genome size. Only CG clusters >3 mutations were considered. Black lines show median values. Source data in S4A and S4C Table. (B) The distribution of the sum of total cluster lengths per yeast and cancer genomes (same as in (A)). Only CG clusters >3 mutations were considered. Black lines show median. Gray circles in (A) and (B) indicate individual yeast isolates or tumor samples. Source data in S4A and S4C Table. (See also S4 Table.) AdenoCA, adenocarcinoma; APOBEC, apolipoprotein B mRNA editing enzyme, catalytic polypeptide-like; CG, C- and/or G-containing; SCC, squamous cell carcinoma; ss, single-stranded; TCC, transitional cell carcinoma; WT, wild type.

indicating that the propensity for cluster formation may differ between different cells repairing DSBs (S4D Table). Since it was possible that clusters found in Can1-Red colonies resulted from the hypermutable subpopulation carrying rare double mutation events, we also sequenced the genomes of other yeast isolates in the population that were not Can1-Red to determine the existence of gamma-induced clusters. Specifically, we analyzed whole genomes from the following diploid colony types: Can1-White single mutant isolates, small colonies with and without rearrangements (based on pulse-field gel electrophoresis [PFGE]; see S1 Fig), and randomly picked colonies of normal size (large). We found that these types of colonies also harbored multiple gamma-induced clustered mutations per genome. For each isolate type, gamma exposure significantly increased the incidence of cluster formation (*P*-values < 0.05, Fisher's exact test; see Fig 2C and 2D and S2D Table for more details and exact *P*-values). Over 20% of the sequenced genomes of each isolate type had at least one nonselected cluster enriched with APOBEC signature mutations. Furthermore, there was no genome-wide induction of C:G to T:A scattered (nonclustered) mutations by A3A expression in G2-arrested cells in the absence of gamma exposure as defined by whole-genome sequencing of Can1-White, unselected small white colonies without rearrangements, and unselected large white colonies, suggesting that the mere presence of A3A in the cell is not sufficient for cluster formation, but rather, it is the presence of persistent ssDNA that is the rate-limiting factor (S2 Fig, S1F Table, S1 Data). Moreover, to eliminate the possibility that gamma exposure itself influenced C:G to T:A mutation loads or cluster formation, we sequenced 29 yeast isolates harboring an empty vector (no A3A) and found no enrichment with APOBEC signature across genomes of these isolates (S2A Table).

DSBs should have been repaired before the first postirradiation cell division because such clusters observed in our isolates were induced in a single G2-arrested gamma-irradiated cell. Interestingly, small colonies with rearrangements had a higher load of C:G to T:A scattered mutations after gamma exposure combined with A3A expression, and, in addition, they also had a higher frequency of clusters per genome than small colonies with no rearrangements from the same experiments. It is possible that these cells may have endured complex molecular events that exposed more ssDNA to damage and/or delayed restoration of ssDNA to dsDNA.

In summary, these data suggest that in the course of repairing bursts of DSBs, ssDNA intermediates are generated which can accumulate multiple DNA lesions and, in turn, can lead to massive formation of clustered mutations in a single cell generation. This outcome is more pronounced in the subpopulation (represented in our experiments by Can1-Red isolates) prone to APOBEC mutagenesis and cluster formation. It appears that the amount of persistent ssDNA dictates the extent to which clusters can form and not an ssDNA-damaging mutagen per se.

## A variety of cluster types suggests multiple pathways of cluster formation

As outlined in Fig 1 and in [5], the strand assignment of mutagenesis in the clusters may inform on mechanisms underlying cluster formation. We categorized clusters containing only

mutated C's and/or G's (referred as CG clusters throughout the paper) into four prominent types (Fig 4A, S2G Table, S3A Table and Materials and Methods): (I) C- or G-coordinated clusters, in which only C's or only G's were mutated on the top (Watson) strand; (II) C-coordinated_G (or G-coordinated_C), i.e., C- or G-coordinated clusters with a single noncoordinated terminal C or G, respectively; (III) CG single-switch clusters in which there was a single switch between the strand-coordinated stretches of mutated C's and G's; and (IV) CG multiple-switch clusters, in which there were alternating mutations in C's and G's in the top strand throughout the cluster. We note that clusters in subcategory (II), C-coordinated_G (or G-coordinated_C), could represent either C- or G-coordinated clusters that colocalize with a single independent C or G mutation at either end of the cluster or single-switch clusters of simultaneous events that had all mutations but one in either C's or G's. We found that in this category of clusters, the distance between a noncoordinated (nonmatched) mutation and the adjacent mutation tends to be greater than the distance separating the last coordinated (matched) mutation from the adjacent mutation. This finding suggests that in some cases, noncoordinated terminal mutation at the end of a true strand-coordinated cluster is there due to random colocalization, while in other cases, a noncoordinated mutation could belong to the cluster generated by a strand-switching mechanism (S3D Fig, S3 Table). Since we could not distinguish between these two possibilities, these clusters were treated separately in our analysis. Finally, to maintain a conservative approach when defining cluster types, we only considered clusters that contained >3 mutations, which provided further statistical confidence, especially for the categorization of single-switch clusters that by this approach had to have at least two consecutive mutated C's followed by at least two consecutive mutated G's or vice versa. This filtering was also applied to analysis of CG clusters in cancers (see below), in which smaller clusters had a greater chance to occur by random colocalization of mutations. However, based on our prior results [12, 17, 24], CG clusters with >3 mutations had higher enrichment with APOBEC mutation signature as compared to nonclustered mutations or to mutations in clusters of smaller size. The numbers of clusters belonging to three major cluster types in Can1-Red isolates from wild-type (WT) diploid yeast were highly correlated, further supporting the idea that there is a balance between different DSB repair pathways that define the formation of a specific category of clusters in cells with differing amounts of long ssDNA.

We determined the distribution of these cluster types among all CG clusters with >3 mutations (Fig 4B, S2H Table). About one-third of the clusters in both diploid and haploid strains were C- or G-coordinated, while single-switch clusters and multiple-switch clusters were about 37% and 11% in haploids and 14% and 44% in diploids, respectively. We also compared the cluster type distributions in the subpopulation forming Can1-Red isolates with cells that do not belong to this subpopulation and determined that although the proportion of multiple-switch clusters varies among these cell types, the ratio between nonselected C- or G-coordinated clusters versus nonselected CG single-switch clusters in Can1-Red colonies and non-Can1-Red strains were not significantly different (S2H Table). One explanation for the large proportion of multiple-switch clusters in diploids as compared to haploids could be that clusters independently formed close to each other on different copies of the homologous chromosomes, thereby appearing as a single cluster based on whole-genome sequencing reads. Another explanation could be that because diploids have two sets of chromosomes, there is higher potential for survival of complex genetic events that could result in multiple-switch clusters. Since these possibilities confound analysis of the mechanisms of cluster formation, we decided to focus on haploids to further assess the molecular mechanisms of ssDNA formation. We chose to analyze cluster formation in Can1-Red isolates because in our experiments, these isolates are coming from a subpopulation of cells with the highest propensity for cluster formation and thus would be the best source of multiple events for analysis of cluster spectra in

A.

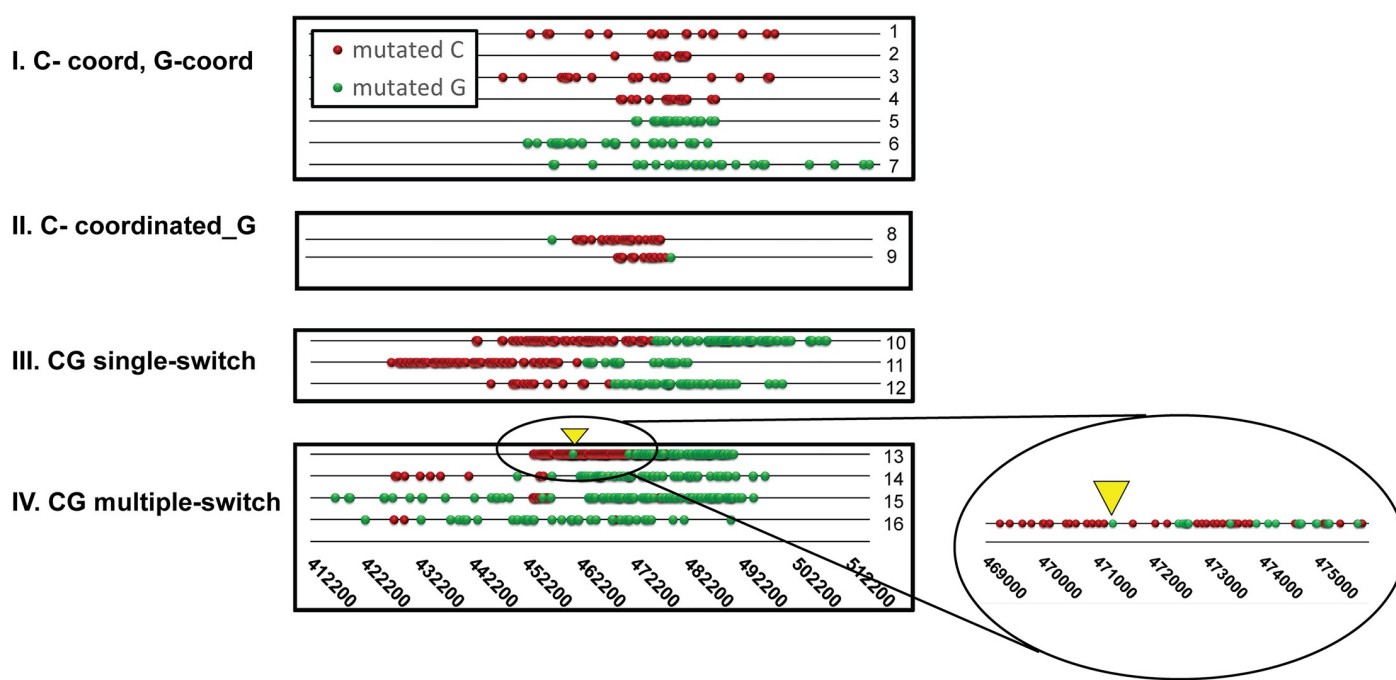

B.

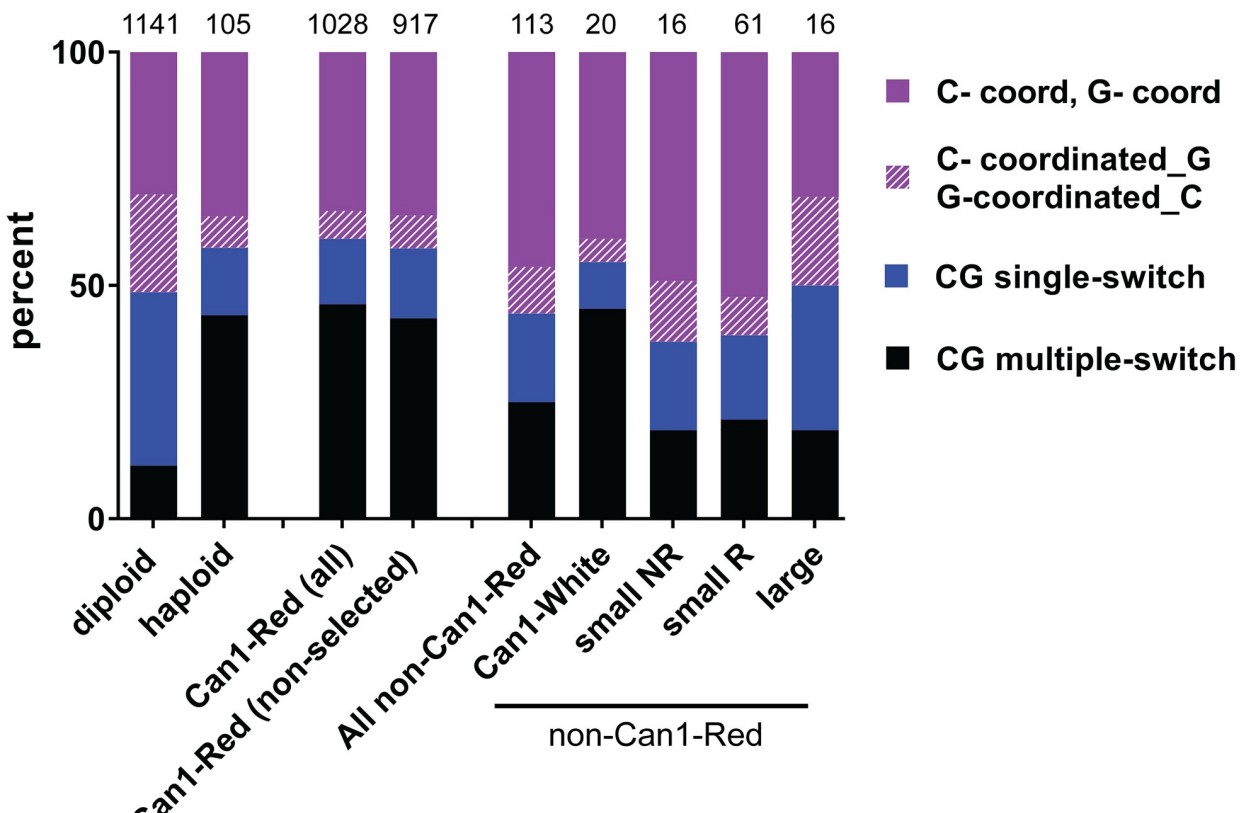

**Fig 4. Cluster types in gamma-exposed yeast.** (A) Example of selected clusters overlapping with the triple reporter on chromosome 2 found in WT Can1-Red diploid or haploid gamma-exposed yeast. Symbols include: "red spheres" = mutated C's, "green spheres" = mutated G's. Each numbered line (1–16) shows a unique cluster. Information about these clusters and their derivative strains can be found in S2G Table and S1 Data. Four cluster types are illustrated: (I) C-coordinated or G-coordinated clusters, comprised exclusively of either mutated C's or mutated G's on the top strand; (II) C-coordinated clusters adjacent to a single G. The terminal G in these clusters may have originated from the same mutagenic event that formed the C-coordinated portion of the cluster or it could be an unlinked random, scattered mutation. Since we could not distinguish between these two possibilities, they are represented here as a separate class. G-coordinated clusters adjacent to a single C were also detected (not illustrated); (III) CG single-switch clusters include clusters that have a single switch in coordination of C to G in the top strand in a 5′ to 3′ direction (called 5′ C to G single-switch) or have a switch from G to C in a 5′ to 3′ direction (called 5′ G to C single-switch) (not illustrated); (IV) CG multiple-switch clusters having alternating mutated C's and G's in the top strand. A portion of cluster number 13 is magnified to illustrate the finer detail of CG switching. Yellow triangle in magnified portion shows corresponding position in unmagnified view. A complete list of cluster categories reflecting strand assignment of mutations is provided in Materials and Methods. (B) The distribution of selected and nonselected gamma-induced cluster types formed from >3 C and/or G mutations in a cluster identified in the following colony types: (i) diploid = selected and nonselected clusters from all WT diploid isolates (Can1-Red and non-Can1-Red isolates); (ii) haploid = selected and nonselected clusters from Can1-Red WT haploid isolates; (iii) Can1-Red all = selected and nonselected clusters from Can1-Red isolates; (iv) Can1-Red (nonselected) = only nonselected clusters from Can1-Red isolates; and (v) non-Can1-Red = clusters from all non-Can1-Red isolates including Can1-White, small NR (not rearranged), small R (rearranged), and large colonies. The four classes of clusters shown are detailed in (A) and include (i) C-coord, G-coord: C- or G-coordinated clusters; (ii) C-coordinated_G, G-coordinated_C: C-coordinated clusters adjacent to a single G or G-coordinated clusters adjacent to a single C, respectively; (iii) CG single-switch; and (iv) CG multiple-switch. Source data in S2H Table. (See also S1, S2 and S3 Tables, S3 Fig). CG, C- and/or G-containing; WT, wild type.

different mutants. Also, it was recently demonstrated that in cultured cancer cell lines, a high frequency of APOBEC mutagenesis and cluster formation also occurs in a subpopulation of cells [20]. This suggests that tumors with multiple mutation clusters also develop from a subpopulation prone to clustered mutagenesis.

## Bidirectional resection during HR repair of bursts of DSBs generates a distinct pattern of clustered mutations

Based on previous studies, long ssDNA formed by 5′ to 3′ resection at site-specific DSBs and at uncapped telomeres is an ideal substrate for cluster formation in conditions of high density of DNA lesions [6, 19, 21]. In addition, 5′ to 3′ resection is a required step in all HR repair pathways (outlined in Fig 1). It can occur as bidirectional resection at DSBs through two stages: first, the initiation stage that requires *SAE2* for short 5′ to 3′ resection prior to strand invasion into a homologous template; and second, long-range resection that can occur via two independent pathways, one requiring *EXO1* and the other *SGS1/DNA2* [25–27]. To test the role of both short- and long-range resection in cluster formation, we repeated gamma irradiation combined with A3A expression in the following resection-defective strains: *sae2Δ*, *exo1Δ*, *sgs1Δ*, and *sgs1Δ exo1Δ*. For all resection-defective strains, the induction of Can1-Red mutants after 40 krad gamma exposure was observed, and the presence of selected and nonselected clusters was confirmed by whole-genome sequencing of multiple isolates (Fig 5A, S1E Table, S2B Table).

Importantly, when gamma-induced Can1-Red frequencies were compared to those of WT strains, single mutants in long-range resection pathways (*sgs1Δ*, *exo1Δ*), as well as the double mutant *sgs1Δ exo1Δ*, were not significantly different from WT strains (S1D Table), suggesting that despite defects in long-range resection, other mechanisms such as BIR may have compensated for cluster formation. Interestingly, when compared to WT, Can1-Red frequencies of *sae2Δ* increased (*P* = 0.0030, two-tailed *t* test, S1D Table) despite having a large reduction in survival rates as compared to WT (median survival for WT = 22%, *sae2Δ* = 5%; see S1B Table for more details).

To further assess the role of long resection in cluster formation, we examined the distribution of cluster types among Can1-Red isolates. Since Sgs1 and Exo1 are known to participate in long-range resection at DSBs, we predicted, based on the switch from top to bottom strand expected at bidirectional resection, that eliminating these pathways would result in a decrease of single-switch clusters with mutated C's on the 5′ end and G's on the 3′ end, reading the top

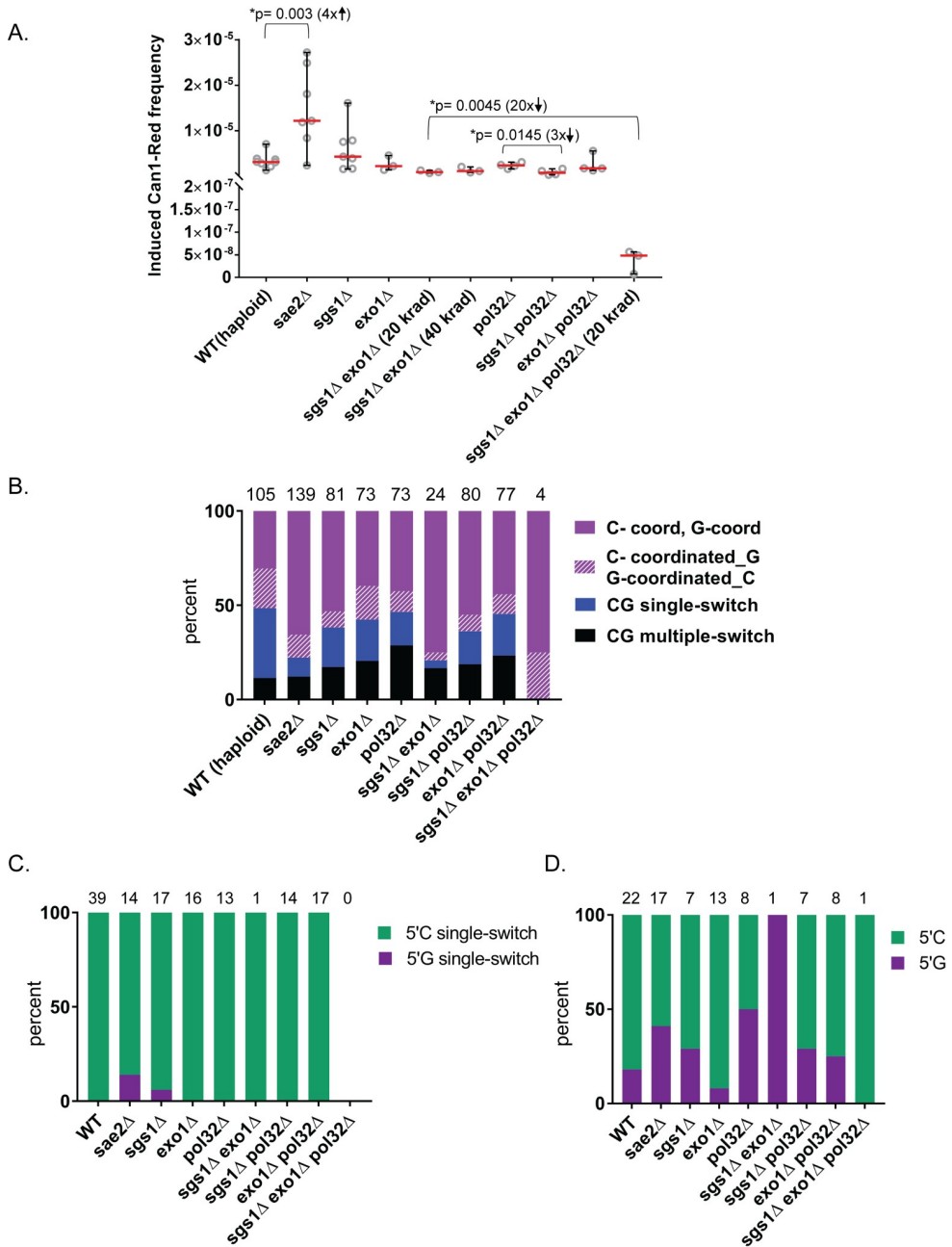

**Fig 5. Cluster analysis in WT and in mutant haploid yeast.** (A) Frequency of gamma-induced Can1-Red mutants in haploid WT, resection-deficient, and BIR-deficient yeast strains (see Fig 1C and text for functional assignment of genetic defects). All yeast strains were exposed to 40 krad gamma irradiation except where noted as 20 krad. Red lines show medians, and black bars show 95% confidence intervals. Significant *P*-values from *t* tests are shown with the corresponding fold difference between medians of Can1-Red gamma-induced frequencies shown in parentheses. Source data in S1E Table. (B) The distribution of selected and nonselected gamma-induced clusters types containing C and/or G >3 mutations in haploid yeast strains. Source data in S2I Table. (C) Percent of CG single-switch clusters with >3 mutations that have either 5′C-3′G (indicated in green) or 5′G-3′C (indicated in purple). Source data in S2I Table. (D) Percent of C-coordinated clusters adjacent to a single G or G-coordinated clusters adjacent to a single C that have either a 5′C (indicated in green) or a 5′G (indicated in purple). Source data in S2I Table. (See also S1, S2 and S3 Tables). BIR, break-induced replication; CG, C- and/or G-containing; WT, wild type.

(Watson) strand (Fig 1B). Indeed, we found an almost complete elimination (1 out of 24 clusters) of single-switch clusters in *sgs1Δ exo1Δ* strains (Fig 5B, *P* = 0.0011, Fisher's exact test; S2E and S2I Table). In addition, single-switch clusters in single mutants *sae2Δ*, *sgs1Δ*, and *exo1Δ* also significantly decreased as compared to WT strains (Fig 5B, *P* < 0.05, Fisher's exact test; S2E and S2I Table). Furthermore, single-switch clusters in all strains, including WT, predominantly conformed to the mutation pattern of 5′C followed by 3′G (as opposed to 5′G followed by 3′C) (Fig 5C, S2I Table). In addition, C- or G-coordinated clusters adjacent to a single C or a single G did not have an obvious pattern of 5′C versus 5′G, further supporting the need to separate this cluster type from others (Fig 5D, S2I Table). Together, these data strongly suggest that single-switch clusters formed in haploids result from bidirectional resection and that long resection plays a critical role in generating this cluster type.

## Long unidirectional resection and BIR pathways contribute to the formation of hypermutable ssDNA and mutation clusters

There was an increase in the contribution of C- or G-coordinated clusters into cluster formation in resection-defective strains as compared to WT strains. Specifically, in the *sae2Δ* mutant and *sgs1Δ exo1Δ* double mutants, the fraction of this cluster type increased significantly (Fig 5B, *P* < 0.001, Fisher's exact test; S2E and S2I Table).

C- or G-coordinated clusters could have resulted from several molecular mechanisms, including asymmetrical resection, i.e., long resection occurring on only one side of the break. In this case, the other side of the break may only partially resect enough to complete repair but not to generate hypermutable ssDNA. In support of asymmetrical resection in resection-defective yeast, we compared the ratios of mutation tracts on the left and right side of single-switch clusters for *sae2Δ*, *sgs1Δ*, and *exo1Δ* mutants and indeed found significantly greater variances in these strains as compared to WT (S4G Table). BIR is another mechanism that can explain C- or G-coordinated cluster formation. Briefly, BIR occurs via the following steps: (i) 5′ to 3′ unidirectional resection at one end of a break (part of the broken chromatid on the other end of a break does not participate in repair and is potentially lost); (ii) 3′ OH invasion of resected DNA end into a homologous template; and (iii) initiation of DNA synthesis via replication bubble with a long ssDNA tail following the bubble (Fig 1A [28, 29]). If BIR was the underlying mechanism for the majority of clusters in *sgs1Δ exo1Δ*, the broken end of the DSB generated in our experiments could be invading a sister chromatid at the identical genomic position, which would not result in copy number variations (CNVs). Invasion into a nonhomologous position, however, would result in copy number increases detectable by read counts. However, only 2 out of 29 clusters in *sgs1Δ exo1Δ* were overlapping with CNVs (S4E and S5A Tables and S5B Fig). In fact, from all the remaining haploid strains, we detected 655 CNVs, yet only one out of these CNVs overlapped with a cluster (S5C Fig). To estimate the contribution of BIR in generating gamma-induced C- or G-coordinated clusters, we explored cluster induction in yeast strains lacking Pol32, a subunit of Pol δ that is not required for S-phase replication but is essential for BIR [30]. Surprisingly, this neither changed the fraction of C- or G-coordinated clusters as compared to WT nor altered the Can1-Red mutation frequency or survival (Fig 5, S1B and S2B Tables). We did, however, find that C- or G-coordinated clusters were significantly longer in *pol32* mutants as compared to WT, though there was no difference in mutation density (S4 Fig, S2J Table). Previous studies in yeast strains in which BIR can be stimulated by a site-specific DSB have shown that in the absence of Pol32, break repair involves long resection at one end of the break [31], which can explain the incidence of C- or G-coordinated clusters found in *pol32Δ* mutants (Fig 1A [ii]). Indeed, the combined role of one-ended resection spanning over dozens of kb and of BIR in the generation of mutation clusters was demonstrated in a

study with a single BIR event stimulated by a site-specific DSB [14]. To test this explanation, we eliminated long resection by deleting *sgs1Δ* and *exo1Δ* in a *pol32Δ* background. The triple mutant had low survival, only 1% after 40 krad and about 6% after 20 krad (the latter is similar to the *sgs1Δ exo1Δ* double mutant) (S1B Table). Therefore, we decided to further assess cluster formation in *sgs1Δ exo1Δ pol32Δ* only after 20 krad exposure.

Importantly, the frequency of Can1-Red induction after 20 krad gamma exposure reduced by about 20-fold in *sgs1Δ exo1Δ pol32Δ* triple-mutant strains as compared to *sgs1Δ exo1Δ*, while there was no significant difference in mutation frequencies in *sgs1Δ pol32Δ* and *exo1Δ pol32Δ* as compared to *sgs1Δ exo1Δ* strains (Fig 5A). Moreover, Can1-Red frequency after gamma irradiation in the triple mutant was at background level, and there was no significant difference between Can1-Red frequency with no exposure versus frequency after exposure (*P* = 0.7589, two-tailed *t* test) (Fig 5A, S1C and S1D Table). This result drastically contrasts with significant gamma induction of Can1-Red in the *sgs1Δ exo1Δ* double mutants (*P* = 0.0098, two-tailed *t* test) (Fig 5A, S1C and S1D Table). Therefore, it is likely that rare clusters found in the genomes of Can1-Red isolates from triple mutants were not gamma-induced. To further assess this, all Can1-Red colonies (10 samples) that occurred after gamma irradiation of A3A-expressing *sgs1Δ exo1Δ pol32Δ* triple mutant were sequenced, and out of these 10 samples, two lost the triple reporter. In contrast, physical loss of the triple reporter was not observed among the 27 *sgs1Δ exo1Δ* Can1-Red colonies isolated after the same gamma dose (20 krad). This suggests that the very small (and not passing statistical test) increase of Can1-Red frequencies caused by gamma irradiation in the triple mutant (Fig 5A) could have resulted in part from gross chromosomal rearrangements that led to the loss of the reporter and were not associated with gamma-induced hypermutable ssDNA tracts. Importantly, in the triple *sgs1Δ exo1Δ pol32Δ* strain, there were only 4 CG clusters with >3 mutations. Moreover, low Can1-Red frequencies support the conclusion that these rare clusters were not gamma-induced but rather likely occurred during growth before DSB gamma induction and thus were not associated with DSB repair. Together, these results strongly suggest that persistent ssDNA during DSB repair is strongly decreased when both BIR and resection are absent and that both processes play independent roles in cluster formation.

## Patterns of clustered mutagenesis in highly APOBEC-enriched cancers are similar to patterns of clusters formed in resection-defective yeast

Using the knowledge gained from our results in yeast about molecular mechanisms contributing to the formation of different cluster types, we next scrutinized mutation catalogs from whole-genome–sequenced tumors belonging to cancer types highly enriched with APOBEC mutagenesis ([9, 12] and Materials and Methods). We hypothesized that, similar to yeast, clustered mutagenesis in APOBEC-enriched cancer types is likely to highlight genome locations where persistent long stretches of ssDNA have occurred. Highly APOBEC-enriched cancer types—including bladder, breast, cervix, head and neck, and lung—had in some tumors over a hundred CG-containing clusters with greater than 3 mutations in a cluster (Fig 6A and 6B, S6B and S6C Table, S1 Data).

Furthermore, the mutation clusters in these cancers were highly enriched with APOBEC signature mutations, indicating that many samples in these cancer types experienced conditions that resulted in multiple ssDNA regions throughout the genome that were vulnerable to hypermutation. We used CG clusters with >3 mutations to calculate the minimum estimate of hypermutable ssDNA formed through the history of a cancer sample (see Materials and Methods) and found that these estimates were close to values observed in yeast. Surprisingly, the estimates of the total amount of ssDNA per genome were within the same range for yeast and

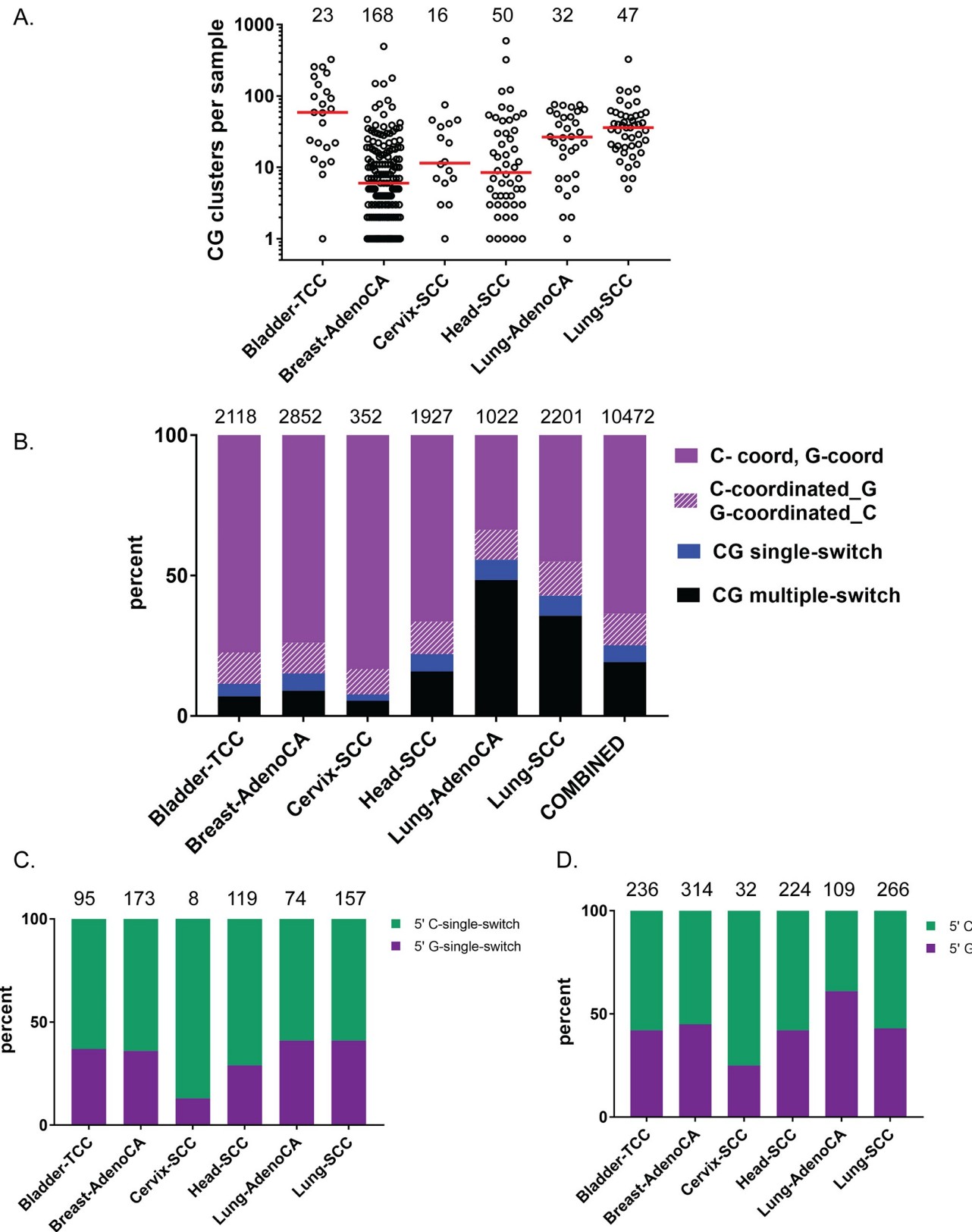

**Fig 6. Cluster analysis in highly APOBEC-enriched cancers.** Mutation clusters described in this figure were derived from the PCAWG data set [32] and were identified similarly to [5, 19]. Analysis uses only CG clusters with >3 mutations. (A) Number of CG clusters containing >3 mutations per

cancer tumor sample in highly APOBEC-enriched cancer types. Black circles indicate individual tumors. Red lines show median values. Numbers above distributions indicate the number of tumor samples per cancer type. Source data in S6B Table. (B) The distribution of CG cluster types with >3 mutations in highly APOBEC-enriched cancers. The four classes of cluster types are described in Fig 4. The numbers above the histogram are the total counts of all CG clusters containing >3 mutations in a cancer type. Source data in S6C Table. (C) Percent of CG single-switch clusters with >3 mutations that have either 5′C-3′G (indicated in green) or 5′G-3′C (indicated in purple). Numbers above histogram are the total counts of all CG single-switch clusters containing >3 mutations in a cancer type. Source data in S6C Table. (D) Percent of C-coordinated clusters adjacent to a single G or G-coordinated clusters adjacent to a single C that have either a 5′C (indicated in green) or a 5′G (indicated in purple). Numbers above histogram are the total counts of C-coordinated adjacent to a single G or G-coordinated adjacent to a single C containing >3 mutations in a cancer type. Source data in S6C Table. (See also S5 and S6 Tables, S3 Fig). AdenoCA, adenocarcinoma; APOBEC, apolipoprotein B mRNA editing enzyme, catalytic polypeptide-like; CG, C- and/or G-containing; PCAWG, Pan Cancer Analysis of Whole Genomes project; SCC, squamous cell carcinoma; TCC, transitional cell carcinoma.

for human cancer genomes (Fig 3B, S4A and S4C Table), while the sizes of these genomes differ by three orders of magnitude (also see Discussion).

Since in yeast, the four categories of CG clusters (Fig 4A) turned useful for suggesting mechanisms underlying hypermutable ssDNA generation, we divided CG clusters in cancers into the same groups (Fig 6B, S5A Table). Like in yeast, category II (containing C- or G-coordinated clusters with a single noncoordinated terminal C or G, respectively) was apparently a mix of true strand-coordinated and single-switch clusters (S3D Fig, S5A and S5B Table) and therefore was analyzed separately. We examined the distribution of all CG cluster types with >3 mutations (over 10,000 clusters in the data set) and found a striking and consistent pattern among all highly APOBEC-enriched cancer types—a high proportion of C- or G-coordinated clusters (34%–83%) and a very low proportion of single-switch clusters (2%–7%) (Fig 6B, S6C Table). This pattern of cluster distribution mimicked that of resection-defective yeast strains *sae2Δ* and *sgs1Δ exo1Δ* (Fig 5B). We further examined the pattern of strand assignment in single-switch clusters and found that unlike yeast, there was no strong bias in favor of 5′C to 3′G versus 5′G to 3′C (compare Fig 6C and Fig 5C). Such a bias is anticipated for single-switch clusters resulting from bidirectional resection of a DSB (Fig 1B). Therefore, we propose that even the single-switch clusters found in these cancers were not formed by bidirectional resection. Single-switch clusters represent a category of clusters that are not indicative of any known mechanism, which is similar to multiple-switch clusters. We found that only lung cancers had a high proportion of multiple-switch clusters, which possibly suggests that complex genetic events in these cancers contributed to cluster formation. The lack of bias of 5′C or 5′G was also found in the subset of coordinated clusters with one terminal noncoordinated C or G (Fig 6D, S6C Table). Together, these analyses suggest that the predominant mode of cluster formation in high-APOBEC cancers may result from a BIR-like mechanism or by some other, yet unknown replication-associated mechanism that leads to C- or G-coordinated clusters via multi-kb replication fork uncoupling.

## Discussion

We have investigated the formation of long stretches of hypermutable ssDNA in eukaryotic yeast cells repairing multiple radiation-induced DSBs. The ssDNA-specific cytidine deaminase APOBEC3A (A3A) expressed at the time of DSB repair left permanent mutational marks at sites of transient ssDNA that was sufficiently persistent for mutagenic damage. These marks of APOBEC mutations occurred in close proximity, resulting in mutation clusters. The frequency of clusters as well as strand assignment of mutations in strains with diagnostic deficiencies in different branches of HR allowed us to highlight subpathways of DSB repair in which long hypermutable ssDNA is formed (Fig 1). DSBs in G2-arrested cells were repaired via HR with a sister chromatid, which is also a prominent DSB repair pathway in S- or G2-phase mammalian cells [33]. Considering the overall conservation of HR DSB repair pathways from yeast to

humans, we compare cluster patterns in a yeast model with APOBEC mutation clusters in human tumors and propose a model of cluster formation in humans.

## Abnormally long persistent ssDNA in yeast is generated by long resection and by BIR in a small fraction of gamma-induced DSBs

Our results indicate that several kinds of abnormal repair of DSBs generated hypermutable ssDNA susceptible to the formation of APOBEC mutation clusters in yeast and that cluster-prone ssDNA generation relied upon pathways of HR repair involving end resection and BIR. We speculate that when a cell is presented with a burst of DSBs, some proteins required for these pathways may be depleted, and repair proceeds abnormally. When both end resection and BIR were blocked in our strains, no gamma-induced clusters were observed (Fig 5 and Results). There was a distinct class of single-switch clusters involving long resection that appeared to occur via bidirectional resection around a DSB (Figs 1B, 4A, 5B and 5C). Clusters in this class indicated that resection tracts can span dozens of kb on either side of the break (Fig 4A [III]), which is much longer than what normally would be expected for allelic gene conversion that generates 2–4 kb of resected DNA [34]. In other studies, longer resection was observed around site-specific DSBs in yeast when a homologous template for HR-driven DSB repair was not available [27, 35] but not in G2-arrested gamma-irradiated yeast [22, 36]. Previously, we have shown that such delayed repair after long-range resection can occur and results in vast mutation clusters when exogenous oligonucleotides are provided to yeast cells six hours after a site-specific DSB has been generated and DNA damage was applied [6].

BIR and/or long-range one-ended resection would generate strand-coordinated APOBEC mutation clusters (Fig 1A). Based on our previous study with a single site-specific DSB [14], a single mutation cluster can result from hypermutable ssDNA formed by both processes. We observed the formation of multiple strand-coordinated clusters caused by bursts of gamma-induced DSBs in strains with a genetic block in one of these two processes. This indicates that either of these pathways can underlie cluster formation in yeast cells challenged with repairing numerous DSBs. It remains to be established however, what the relative contributions of these processes in long ssDNA formation during repair of numerous gamma-induced breaks in WT yeast are.

The density of APOBEC-induced mutations in clusters (approximately 1 mutation/kb) was close to the density observed in artificially created yeast subtelomeric ssDNA exposed to endogenously expressed A3A for up to 48 hours [19]. This suggests that the amount of long persistent ssDNA, rather than the level of endogenously expressed A3A, was a rate-limiting factor for cluster formation in gamma-irradiated yeast. Previously, others have shown that the bulk size of ssDNA formed in resection tracts observed in G2-arrested WT yeasts repairing gamma-induced DSBs was only around 1–2 kb and that resection was mostly symmetrical in WT cells [22, 23]. Thus, APOBEC clusters in our experiments could have been formed from a small fraction of DSBs in which repair was significantly delayed, allowing formation of unusually long persistent ssDNA by extensively long bidirectional resection, unidirectional resection, and BIR. We estimate that the fraction of such abnormally repaired DSBs was around 5% in WT haploid yeast cells (120 clusters per 47 whole-genome–sequenced haploid WT strain that survived after repairing 50 gamma-induced DSBs). Further studies are needed to understand the cause(s) of delayed repair of gamma-induced DSBs in G2 cells where sister chromatid should be immediately available for HR.

## Pathways of APOBEC mutation cluster generation operating in human cancers

Transcription, as well as DSB repair and DNA replication, involves transient ssDNA intermediates. However, in human cancers, mutagenesis associated with ssDNA-specific APOBEC

enzymes was observed to be equally distributed in the transcribed and non-transcribed strands of genes [37]. Both clustered and scattered APOBEC mutagenesis in cancers were more prominent in early-replicating regions of cancer genomes, and this preference was similar in transcribed and non-transcribed regions [24]. Thus, transient ssDNA formed at transcription is not a likely source of mutation clusters in human cancers. Normal eukaryotic DNA replication involves formation of only a hundred nucleotides of ssDNA in lagging strand, which can increase to around a thousand nucleotides in genotoxic stress conditions [38]. This is much shorter than the dozens of kb of ssDNA necessary to account for clusters formed in cancers.

Based on size (i.e., number of mutations in a cluster), length, and mutation density, many APOBEC clusters in human cancers (Fig 3 and S3 Fig) resemble clusters formed in yeast cells experiencing repair of multiple simultaneous DSBs. Most 5′ to 3′ resection in mammalian cells measured at site-specific DSBs extended as far as 3,500 nt [39]. This is in agreement with estimates of an average resection length of 3.4 kb from multiple DSBs induced by expression of restriction endonucleases; however, longer resection tracts (up to 13 kb) were also observed in Lig4−/− p53−/− cells [40]. Relatively short (less than 2 kb) 5′ overhangs as well as 3′ overhangs were also documented by sequencing processed hairpin-capped ends of DSBs generated by V (D)J recombination RAG endonuclease [41]. DSB resection in mammalian cells strongly depends on CtIP (Sae2), MRN (MRX) complex, Exo1, and Dna2 proteins whose homologues define 5′ to 3′ resection in yeast ([39], also reviewed in [42]), suggesting that most DSB end resection in mammalian cells is 5′- to 3′-directed. Our results obtained in a yeast model indicated that symmetrical resection can frequently generate vast APOBEC mutation clusters identified as "single-switch" (Figs 1B, 4 and 5B). In agreement with bidirectional resection going in a 5′ to 3′ direction on each side of the break, the left part of the single-switch clusters contained only mutated C's and the right part only mutated G's in the "top" strand (Fig 1B and Fig 5C). Single-switch clusters with the opposite order of mutated nucleotides (G's on the left, C's on the right side of the switching point) were absent or very rare in yeast (Fig 5C). In contrast, most APOBEC clusters in cancers were completely strand-coordinated (C-only or G-only). There was only a small fraction of single-switch APOBEC clusters in cancers, and many of these single-switch clusters had a G-stretch on the left and thus were inconsistent with 5′ to 3′ direction of bidirectional resection (Fig 6B and 6C). Altogether, comparison between APOBEC mutation in cancers with APOBEC clusters formed in yeast performing HR repair of multiple gamma-induced DSBs suggests that bidirectional resection at DSBs plays essentially no role in the generation of clusters in cancers. We propose that single-switch and multiple-switch clusters in cancer, as well as multiple-switch clusters in yeast, originate from complex repair events involving multiple strand invasions caused by a single initiating DSB [43, 44].

While the bulk of ssDNA tracks formed during mammalian replication and DSB repair are smaller than many APOBEC mutation clusters in cancers, there could be a small fraction of unusually long persistent ssDNA formed by either resection or abnormal replication. In mammalian cells, including cancers, HR is involved in repair of DSBs as well as in recovery of collapsed replication forks [3, 45, 46]. Many kinds of HR-dependent DSB repair may involve a source of long ssDNA—BIR, in which leading and lagging strands are profoundly uncoupled (reviewed in [47]). We propose that the prevailing class of completely strand-coordinated APOBEC clusters in cancers originates from BIR and/or unidirectional resection. However, since long-range resection spanning dozens of kb has not been demonstrated in human cells, a BIR-like mechanism could well be the major pathway generating strand-coordinated clusters. This unusual form of DNA replication can be also a source of chromosome rearrangements. Interestingly, both chromosome rearrangements and APOBEC mutation clusters in cancers occur more frequently in early-replicating regions of the genome [24, 48, 49].

## Minimum estimates of hypermutable ssDNA amounts formed in yeasts repairing multiple simultaneous DSBs and in cancers are comparable

We used the total sizes of APOBEC-induced mutation clusters in yeast genome and total size of APOBEC-enriched mutation clusters in cancers as minimum estimates of amounts of hypermutable ssDNA formed in these genomes (Fig 3B). We report here that the total length of long hypermutable ssDNA accumulated throughout the history of individual tumors, estimated by totaling APOBEC cluster lengths, is comparable with the estimate of total length of ssDNA formed during a single cell generation in the genomes of diploid yeasts repairing bursts of gamma-induced DSBs (Fig 3B, S4A and S4C Table). This is surprising because human genomic DNA is 260-fold longer than that of the yeast genome. While multiple explanations could be offered for such a striking similarity, we would like to propose a speculation that the amount of simultaneously formed long ssDNA is correlated with the number of DSBs that are abnormally repaired or that escape repair. For example, abnormal repair and/or breakage could be due to depletion in RPA, an ssDNA-binding protein required for DNA replication and for DSB repair [50–52]. The proposed correlation between ssDNA and breakage could be due to a well-established phenomenon of ssDNA formation during DSB repair. It could also be due to long and persistent hypermutable ssDNA being more prone to breakage than dsDNA. It is a common view in the field that a single or a small number of unrepaired DSBs can cause cell death [53], and this lethal effect occurs regardless of the genome size [54]. This agrees with our interpretation that the tolerance of hypermutable ssDNA in either yeast or in cancers is defined by the increased chance of forming unrepairable and hence lethal DSBs. If this chance is defined by the total size of ssDNA regions, it should not depend upon the vastly different genome sizes of yeast and human cells. The similarity in toleration of long ssDNA between yeasts repairing a hundred simultaneous DSBs and cancers indicates that there could have been a burst(s) of long ssDNA and/or DSB formation at some point during the history of a tumor. One possible source of explosive formation of chromosome breakage and APOBEC mutation clusters was recently discovered in cultured human cells experiencing telomere protection crisis [10]. Other sources of explosive formation of DSBs in the history of tumors may include replication stress or DNA-damaging anticancer therapies. Episodic bursts of APOBEC mutagenesis that could be associated with bursts of ssDNA formation were recently reported with in vitro-propagated cancer cell lines [20]. It remains to be established whether such bursts of ssDNA formation do occur in the history of tumors and how simultaneous cluster formation is in cancer genomes.

## Long ssDNA prone to hypermutation in cancers: Origins and implications

APOBEC mutation clusters in cancers identify stretches of long persistent ssDNA transiently formed in the proliferative history of a tumor. Vast cluster formation may indicate problems with DNA replication or excessive DNA damage in this tumor. Further studies may show whether cluster formation can be combined with other molecular phenotypes for evaluating prognosis and treatment strategies in cancer. The formation of an APOBEC cluster requires a coincidence of abnormally long ssDNA intermediate prone to damage-induced hypermutation and the exposure of ssDNA to APOBEC cytidine deamination activity. In many tumors of several cancer types used in our analysis (S6A Table), APOBEC activity in damaging chromosomal DNA is sufficiently high to be detected outside mutation clusters on the whole-genome or even on the whole-exome level. However, in cancer types with low APOBEC mutagenesis, the APOBEC mutation signature can be detected mostly or even exclusively in clusters [5, 9, 12, 17, 55]. This underscores the sensitivity of the approach of detecting long ssDNA by mutation clusters left at positions where such ssDNA once was transiently present. Based on our

prior studies in yeast, long persistent ssDNA will be prone to hypermutation caused by DNA lesions other than APOBEC [5, 6, 21, 56] because the lesions cannot be accurately repaired by excision repair pathways relying on an undamaged complementary strand as a template. These non-APOBEC lesions may be one of the sources of clusters not associated with the APOBEC mutation signature [5, 57]. In the absence of acute DNA damage, the density of non-APOBEC lesions may not be sufficient to generate a mutation cluster. However, since hypermutation in ssDNA can severely (up to 1,000-fold) exceed mutation rates in dsDNA, ssDNA-associated damage-induced mutagenesis could still be a significant contributor to genome-wide mutation loads in cancers. Our study points to BIR (or other irregular forms of DNA replication) as a potential source of hypermutable ssDNA in human cancers. Since the formation of ssDNA in mammalian BIR has not yet been evaluated, it becomes important to address this question in models similar to ones explored in yeast (reviewed in [47]).

## Materials and methods

### Yeast strains and plasmids

All diploid and haploid yeast *S. cerevisiae* used in this study were derivatives of ySR128 (*MATalpha ura3Δ can1Δ ade2Δ his7-2 leu2-3,112 trp1-289 lys2::ADE2-URA3-CAN1*), which were made *ung1Δ* by disruption of UNG1 with *KANMX* cassette, conferring resistance to geneticin [5]. Diploid yeasts (MATa/MATalpha) were generated by mating-type switching using the YEpHO-*LEU2* [58]. Strains contained a triple-gene mutation reporter, *CAN1-ADE2-URA3*, that was inserted at the native *LYS2* on the right arm of chromosome II, approximately 230 kb from the centromere and 340 kb from the telomere (Fig 2A). Strains were either transformed with vector pySR435 carrying APOBEC3A (A3A) or vector pySR419 lacking A3A (referred to as empty vector) [16]. Confirmation that the vector did not integrate into the genome but remained autonomously replicating was done by observing the frequent loss of Hyg$^R$ plasmid marker after propagation on the media without antibiotic. Strains containing *sae2*::NATMX, *sgs1*::NATMX, and *pol32*::NATMX disruptions were constructed by transformation with a PCR-derived NATMX cassette (conferring resistance to nourseothricin) containing approximately 80 bp of sequence flanking the 5′ and 3′ regions of desired ORF that was replaced by NATMX via a one-step integration. Strains containing *exo1*::BSD were constructed via a PCR-derived blasticidin (BSD) marker (TEF/BSD; Invitrogen, Carlsbad, CA, USA) containing approximately 80 bp of sequence flanking the 5′ and 3′ regions of *EXO1* that was replaced by BSD via a one-step integration. Replacement of *SAE2*, *SGS1*, and *EXO1* was verified by PCR (primers listed in S7 Table). Strains containing *pol32*::KANMX were constructed by crossing *MATa pol32*::KANMX strains (yCS177 or yCS178) with *MATalpha ung1*::KANMX *sgs1*::NATMX *exo1*::BSD strain (yCS124 or yCS125), followed by tetrad dissection and confirmation of haploid strains with desired genotype by PCR and phenotypic analysis. All strains are listed in S8 Table.

### A3A mutagenesis in yeasts repairing gamma-induced DSBs

Yeast strains were transformed with pySR435 A3A-expression vector (or pySR419 empty vector) containing an Hyg$^R$ cassette (conferring resistance to hygromycin), and transformants were purified by streaking for single colonies onto YPDA + Hyg plates (200 μg/mL hygromycin). Several single colonies were immediately frozen in 20% glycerol at −80˚C. For every experiment, yeast from the frozen stock of a transformant colony was streaked for single colonies, and a single colony was patched on YPDA + Hyg and allowed only minimal growth to reduce accumulation of background mutagenesis by A3A before gamma irradiation. The yeast patch was then used to inoculate 5 ml of YPDA liquid medium containing 100 μg/mL of hygromycin and grown in a shaker (200 rpm) at 30˚C for 8–10 hours. This suspension was

then diluted by fresh YPGA + Hyg to a cell titer of about $5 \times 10^3$ cells/mL, and 250 ml of this suspension was grown for approximately 15 hours in a 2-L flask to mid-log phase (a titer of $5 \times 10^6$ cells/ml). Cells were then arrested at G2 by adding nocodazole to cultures at 30˚C with continued shaking, first at a concentration of 0.02 mg/mL, followed by two subsequent boosts with nocodazole every hour at a concentration of 0.01 mg/mL [59]. Along with the last addition of nocodazole, 0.01 mg/mL doxycycline was also added to the culture to stimulate expression of A3A, so the last hour of incubation was in the presence of nocodazole and doxycycline. After this, the culture was split into the following portions: (i) 10 mL was spun down and frozen at −80˚C in 20% glycerol; (ii) 190 ml was spun down and resuspended in approximately 2 mL of 1 M sorbitol + 0.1 M EDTA, split into three 1 mL portions, and stored at −20˚C for genomic DNA preparation. WGS of these DNAs was used to filter out mutations induced by A3A before gamma irradiation (see below). The remaining 50 ml of G2-arrested culture was boosted by an additional 0.01 mg/mL of nocodazole, split into two 20 mL portions, and placed on ice. One portion was gamma-irradiated using the Shepherd Irradiator Model 431 (San Fernando, CA, USA) with a $^{137}$Cs source at a dose rate of 2 krad/minute, and the other portion was not irradiated and used as a control. After every 10 krad of irradiation, the cell suspensions were aerated by 10 seconds of vortexing. After gamma exposure, both portions of cultures (gamma-exposed and non-gamma–exposed) were maintained at G2 arrest by nocodazole (0.01 mg/mL) and held at 30˚C with shaking for 3 hours. Yeasts were then diluted to appropriate concentrations and plated on complete synthetic medium with low (15 μg/ml) concentration of adenine (Com-Low-Ade), allowing better detection of red Ade⁻ colonies to determine survival. Cells were also concentrated and plated onto synthetic medium with low adenine and lacking arginine but supplemented with 60 mg/L canavanine (Can-Low-Ade) to determine frequencies and to isolate single Can$^R$ and double Can$^R$ Red mutant frequencies. A Can$^R$ phenotype indicates a mutation inactivating the *CAN1* gene, and a red color indicates a likely mutation in *ADE2*. This double-mutant phenotype was designated as Can1-Red. Yeasts on Com-Low-Ade plates were grown at 30˚C for 3–4 days, and yeasts on Can-Low-Ade plates were grown at 30˚C for at least 7 days to allow red color formation. Since Can1-Red mutants had coincident inactivation of the closely spaced *CAN1* and *ADE2* genes, they were expected to have at least one mutation cluster that overlapped the triple reporter region, *CAN1-ADE2-URA3*. This mutation cluster is referred to as a "selected cluster." Clusters that did not intersect with this reporter are referred to as "nonselected clusters" and were often used in analyses because they contained mutations that arose independent of selection.

## Yeast whole-genome sequencing

Yeast colonies for WGS were collected from one of the following media: (i) from YPDA or from Com-Low-Ade plates, where no selection for mutations was applied; or (ii) from Can-Low-Ade plates to select isolates with single Can1 and with double Can1⁻Red mutant phenotypes. Colonies from these media were directly patched on YPDA plates and grown at 30˚C for 2–3 days to allow for the loss of a plasmid with Hyg$^R$ (A3A expression vector or empty vector). Yeast from the patch was suspended in $H_2O$ and spread onto YPDA plates to obtain approximately 200 isolated single colonies. Colonies were screened for sensitivity to hygromycin by replica plating on rich media containing hygromycin to identify yeast that lost the A3A expression vector or the empty vector. Growth on Yeast extract Peptone Glycerol (YPG) plates was also tested to avoid accidentally picking respiratory-deficient isolates, which routinely occur at propagation of yeast strains. Hyg$^R$ YPG+ colonies were then isolated and prepared for whole-genome sequencing using YeaStar Genomic DNA Kit (Zymo Research, Irvine, CA, USA). The chromosomal sizes of small colonies collected from YPDA plates were analyzed by

PFGE similar to [59] to detect large chromosomal rearrangements (see example in S1 Fig). We chose to analyze isolates with rearranged chromosomes separately from isolates without rearranged chromosomes to discern any differences in cluster characterization for isolates that experienced complex genetic events leading to large rearranged chromosomes.

Whole genomes of yeast isolates were sequenced using an Illumina HiSeq 4000 platform (Illumina, San Diego, CA, USA) with >50× coverage. Using CLC Genomics Workbench (QIAGEN, Redwood City, CA, USA), paired-end sequencing reads were mapped to reference genome ySR128. Default parameters were used for initial read alignment, followed by local realignment of mapped reads and removal of duplicate reads. For mutant yeast isolates, the fixed ploidy parameter was used for variant detection with a required variant probability of 99.5%. Marginal variant calls with a minimum frequency of 20% were then filtered. Lowfrequency variant detection with a required significance of 1% was used for pre-gamma–exposed yeast populations, and these mutation calls were used to filter out common, preexisting mutations in mutant yeast isolates using the "compare variants within group" function in CLC Genomics Workbench. Only single-nucleotide base substitution variants (SNVs) were considered in cluster analysis as described below.

## Whole-genome–sequenced human cancers used for APOBEC mutation cluster analysis

We used mutation catalogs of whole-genome–sequenced human solid tumor samples developed by the Pan Cancer Analysis of Whole Genomes project (PCAWG) [32] to compare patterns of APOBEC-induced clusters in cancers and in yeast. Clusters containing only mutations in C and/or G were used in the analysis. In our experiments, yeast formed mutation clusters in conditions of high level of genome-wide APOBEC mutagenesis. Therefore, we chose for comparison the following six cancer types known to be highly enriched with the APOBEC mutation signature [9, 12]: transitional cell carcinoma of bladder (Bladder-TCC), breast adenocarcinoma (Breast-AdenoCA), squamous cell carcinoma of cervix (Cervix-SCC), head and neck squamous cell carcinoma (Head-SCC), lung adenocarcinoma (Lung-AdenoCA), and lung squamous cell carcinoma (Lung-SCC). Enrichment with APOBEC mutation signature in clustered mutations of these cancer types always exceeded enrichment in the catalog of total mutations or in scattered (not belonging to any cluster) mutations (S6A Table). Most of these tumors (336 of 376) contained APOBEC-enriched clusters of multiple C and/or G mutations, in some cases nearly a thousand mutations per cluster (Fig 6A and S5 and S6A Tables).

## Mutation cluster and signature analyses in yeast and in cancer

Mutation cluster analysis was performed as previously described [12, 19]. Briefly, clusters were identified among groups of closely spaced mutations with intramutational distances between 10 bp and 10 kb that were too dense to be due to random variation of intermutational distances. To identify clusters that were unlikely to have formed by random distribution of mutations within a genome, we computed a P-value for each group. Let $x$ = number of bases spanned by a group (from first mutation to last), $k$ = number of mutations in a group, $\pi$ = number of total mutations divided by number of total bases in a genome, and $j$ = an indexing parameter. A cluster P-value was defined as the probability of observing $k - 1$ mutations in $x - 1$ or fewer bp using a negative binomial distribution as follows:

$$p = \sum_{j=0}^{x-k} \binom{k+j-2}{j}(1-\pi)^j \pi^{k-1}.$$

Each group with *P*-values $\leq 10^{-4}$ was considered a bona fide mutation cluster. Mutation groups with <10 bp intermutational distances were considered as a complex event and counted as one mutation because they could have originated from a single mutagenic event rather than from independent DNA lesions.

Mutation clusters containing more than 3 mutations were used as a proxy for long persistent regions of ssDNA because these clusters always showed increased enrichment with APO-BEC mutation signature in cancers [12, 17, 24, 55]. The length of a hypermutable ssDNA region was estimated as the distance between mutations bordering a cluster. These clusters were categorized by nucleotides mutated in the top (Watson) strand as follows: C-coordinated, only cytosines mutated; G-coordinated, only guanines mutated; C-coordinated with a terminal G, a single G mutated either on the 5′-side or on the 3′-side of a cluster; G-coordinated with a terminal C, a single C mutated either on the 5′-side or on the 3′-side of a cluster; CG single-switch, contiguous stretch of at least two mutated G's next to a contiguous stretch of at least two mutated C's (in some analyses, this category was divided into two subcategories: 5′ C single-switch clusters, mutated C-stretch on the 5′-side followed by a mutated G-stretch; and 5′ G single-switch clusters, mutated G-stretch on 5′-side followed by a mutated C-stretch); and CG multiple-switch, clusters with mutated G's and C's with more than one contiguous stretch of mutated C's and/or mutated G's.

Enrichments with tCw generic APOBEC signature motif, as well as A3A-like ytCa or APOBEC3B (A3B)-like rtCa signature motifs (mutated base capitalized; y = C or T; r = A or G; w = A or T), across individual yeast or cancer genomes or in mutation clusters of a given genome were calculated and statistically evaluated as previously described [12, 19, 60]. An example of the calculation for enrichment of mutations within ytCa → ytTa signature expected for A3A mutagenesis in *ung1*Δ yeast is presented below, where the context is derived from a 41-nucleotides region containing the mutated residue in the center:

$$Enrichment_{(ytCa \rightarrow ytTa)} = \frac{mutations_{(ytCa \rightarrow ytTa)} \times context_{(c)}}{mutations_{(C \rightarrow T)} \times context_{(ytCa)}}.$$

For each motif, the reverse complement was also used in the calculations. The use of nucleotide context immediately surrounding the mutation rather than the whole genome helped to account for the localized preference of mutagenesis stemming from small-range scanning by mutagenic enzymes such as APOBEC or the preference of lesion occurrence and lesion repair, as well as other factors, including epigenomic features influencing mutagenesis within localized genomic regions [61, 62].

### Identification of CNVs in yeast isolates

To identify CNVs in isolates of the gamma-treated yeast strains, we used BWA-MEM-0.7.10 to align the whole-genome sequencing reads to the reference genome [63]. Duplicate reads were removed from the alignment files via MarkDuplicates from Picard Tools (http://broadinstitute.github.io/picard/). Copy number changes in the strains were annotated using VarScan2 copy number on the pileup files generated using the Samtools mpileup command [64–66]. Only CNVs that were unique to the Can1-Red isolates were detected. We calculated the average depth of coverage across each bam file with Samtools depth. The difference between the depth of the strains being compared was used to inform the data ratio parameter in VarScan2. Circular binary segmentation using the DNAcopy package in R [67] was applied to the output produced by VarScan2. The CNVs that overlap with regions containing clustered mutations were identified using the GenomicRanges software in R [68]. CNVs that overlap with clusters were verified by visual examination of coverage in CLC Genomics Workbench.

CNVs that were not visible in CLC Genomics Workbench were deemed as artifactual calls by VarScan and were removed from analysis. These include focal amplifications for samples 102_3 and 99_11 on chromosome 12 at regions of rDNA.

## Supporting information

**S1 Fig. Chromosomal rearrangements in small WT diploid colonies.** Examples of small WT diploid colonies with chromosomal rearrangements detected by PFGE. Lane 1 shows control strain with no rearrangements. Lanes 2, 4, and 7–12 show isolates with rearranged chromosomes as indicated by orange arrows. Rearranged chromosomes have positions that deviate from those in the control strain. Chromosome 1–16 positions are indicated adjacent to gel image. PFGE, pulse-field gel electrophoresis; WT, wild type.
(TIF)

**S2 Fig. Distribution of scattered and clustered APOBEC mutations in WT diploid yeast.** (A) The total numbers of scattered C:G to T:A mutations per isolate in WT diploid yeast. Source data in S1F Table. (B) The total numbers of clustered C:G to T:A mutations (from selected and non-selected clusters) per isolate in WT diploid yeast. Source data in S1F Table. Gray circles indicate an individual isolate, and red lines show median values. NG, no gamma exposure; G, gamma exposure (80 krad); NR, isolates with no rearranged chromosomes; small, small colonies; small_combined, small colonies with and without rearrangements combined; large, large colonies. *P*-values were calculated from Mann–Whitney *t* test shown above groups. "*" indicates significant *P*-values (<0.05). APOBEC, apolipoprotein B mRNA editing enzyme, catalytic polypeptide-like; WT, wild type.
(PDF)

**S3 Fig. Mutation density, size, and length of clusters in yeast and cancer.** For all graphs in this figure, red lines show median values and black circles show individual clusters in haploid yeast and in diploid yeast (WT strains, nonselected clusters in panels A, B, and C and all clusters in all genotypes in panel D) or clusters in tumor samples. Only CG clusters with >3 mutations were used to make graphs. Clusters are divided into four main types: (i) CG coord, C- or G-coordinated; (ii) coord with terminal CG, C-coordinated clusters adjacent to a single G or G-coordinated clusters adjacent to a single C; (iii) single-switch clusters; and (iv) multiple-switch clusters. Details about cluster types are in Fig 4. (A) Mutation density of clustered mutations. Source data are in "Mutation_Density_per_kb" column of S3A Table for yeast and in "Mutation_Density_per_kb" column of S5A Table for cancer (see S3 and S5 Tables descriptions for details). (B) Size of clusters, i.e., the number of mutations in the cluster. Source data are in "Cluster_Size_Complexes" column of S3A Table for yeast and of S5A Table for cancer. (See S3 and S5 Tables descriptions for details.) (C) Cluster length from the position of the first nucleotide to the last nucleotide in the mutation cluster; reported in bps. Source data are in "Cluster_Length" column of S3A Table for yeast and of S5A Table for cancer. (See S3 and S5 Tables descriptions for details). (D) Distance between the terminal matched or nonmatched residues and the preceding mutated residue in each cluster designated as "coord with terminal CG." "Matched" and "nonmatched" correspond to the columns "Distance_terminal_non_matched" and "Distance_terminal_matched" in the S3 Table and S5 Table for yeasts and cancers, respectively. *P*-values depicted over the graphs were calculated using a two-tailed Mann–Whitney test. Source data in S3B Table and in S5B Table. CG, C- and/or G-containing; WT, wild type.
(PDF)

**S4 Fig. Cluster length and mutation density of C- or G-coordinated clusters observed in haploid mutant and WT strains.** (A) The distribution of cluster lengths for C- or G-coordinated with clusters >3 mutations in haploid strains. Source data in S2J Table. (B) The

distribution of mutation density of clusters C- or G-coordinated with clusters >3 mutations in haploid strains. For all graphs, gray circles indicate individual clusters, and black lines show median values. Median values are written above each distribution. Mann–Whitney two-tailed *t* test with Bonferroni correction for multiple hypothesis testing showed a significant increase in cluster lengths in pol32 mutants as compared to WT, and *P*-value is shown on graph (see S2J Table for more details). There were no significant differences between clusters in *sgs1Δ exo1Δ* from 20 krad versus 40 krad irradiation, and thus, clusters in this strain were pooled together. Source data in S2J Table. WT, wild type.
(PDF)

**S5 Fig. Examples of clustered mutations overlapping with CNVs in the yeast genome and distribution of clusters across yeast chromosomes.** (A) Coverage maps of sequencing reads in *sgs1Δ exo1Δ* yeast isolate (164_sgsexo_28) showing a copy number increase from coordinates 524,672 to 601,156 in chromosome 14, with an overlapping cluster shown as a red horizontal line above the graph. Cluster details are shown below the graph. (B) Coverage maps of sequencing reads in *sgs1Δ exo1Δ* yeast isolate (165_sgsexo_4) showing a copy number increase from coordinates 1,328 to 136,201 in chromosome 8 with an overlapping cluster shown as a red horizontal line above the graph. Cluster details are shown below the graph. (C) Coverage maps of sequencing reads in *sgs1Δ pol32Δ* yeast isolate (227_sgs1pol32_8) showing a copy number increase from coordinates 6,208 to 136,761 in chromosome 5 with an overlapping cluster shown as a red horizontal line above the graph. Cluster details are shown below the graph. Complete list of all CNVs detected in haploid isolates as well as of mutation clusters overlapping with CNVs is in S4E Table. CNV, copy number variation.
(PDF)

**S6 Fig. Distribution of clusters across yeast chromosomes.** The distribution of nonselected clusters across all 16 yeast chromosomes in WT diploid yeast. *P*-values from linear regression analysis shows that clusters are spread across yeast chromosomes proportionately to the size of the chromosomes ($P < 0.0001$). Source data for this figure are shown in S4F Table. WT, wild type.
(PDF)

**S1 Table. Survival and mutation frequencies after gamma irradiation.** Datasheets in S1 Table: S1A Table, survival and mutagenesis after gamma irradiation in diploid yeast; S1B Table, survival and mutagenesis after gamma irradiation in haploid yeast; S1C Table, distribution of Can1-Red frequency in haploid and diploid yeast with and without gamma exposure. Source data for Fig 2A. Values in "Can1-Red Mutation Frequency" column for each group on the scatterplot can be obtained by filtering in the following columns: [Strain = WT], [Ploidy = diploid or Ploidy = haploid], [Gamma exposure (krad) = 80 for diploid or Gamma exposure (krad) = 40 for haploid or Gamma exposure (krad) = 0 for nonirradiated diploids and haploids]. S1D Table, statistical comparisons of induced Can1-Red frequencies; S1E Table, induced Can1-Red frequencies in haploid yeast. Source data for Fig 5A. Values in "Can1-Red Mutation Frequency" column for each of the four groups on the scatterplot can be obtained by filtering in the following columns: [Strain = genotype of the group as indicated on x-axis of scatterplot], [Gamma exposure (krad) = 40, if not indicated on the scatterplot or as indicated for the corresponding group on x-axis of the scatterplot]. S1F Table, CG to TA scattered and clustered mutations in WT diploid yeast with gamma and no gamma exposure. Source data for S2A Fig. Values in "sum_GC_to_TA_scattered_mutations" column for the scatterplot can be obtained by filtering in the following columns for each group: Can1-White NG, [Gamma exposure (krad) = 0], [Isolate type = Can1-White]; Can1-White G, [Gamma exposure (krad) =

80], [Isolate type = Can1-White]; small NG, [Gamma exposure (krad) = 0], [Isolate type = small]; small_NR, [Gamma exposure (krad) = 80], [Isolate type = small_NotRearranged]; small_R, [Gamma exposure (krad) = 80], [Isolate type = small_rearranged]; small_combined G, [Gamma exposure (krad) = 80], [Isolate type = small_NotRearranged or small_rearranged]; large NG, [Gamma exposure (krad) = 0], [Isolate type = large]; large G, [Gamma exposure (krad) = 80], [Isolate type = large]; Can1-Red G, [Gamma exposure (krad) = 80], [Isolate type = Can1-Red]. Source data for S2B Fig. Values in "sum_GC_to_TA_clustered_mutations" column for the scatterplot can be obtained by filtering the values in this column as described for the same groups on S2A Fig. CG, C- and/or G-containing; G, gamma-exposed; NG, non-gamma–exposed; WT, wild type.
(XLSX)

**S2 Table. Cluster details per isolate type.** Datasheets in S2 Table: S2A Table, sum cluster details per isolate type in diploid yeast; S2B Table, sum cluster details per isolate type in haploid yeast; S2C Table, individual yeast isolate cluster details and summary table for number of samples with clusters. Source data for Fig 2C. Values for each category on the bar graph are in the column "Percent samples ≥1 cluster" of the small imbedded table. Source data for Fig 2D. Values in "Total clusters" column for the scatterplot can be obtained by filtering in the following columns for each group. Haploid: Haploid Can1-Red G, [Strain = WT Haploid], [Gamma exposure (krad) = 40], [Isolate type = Can1-Red]; Diploid: Can1-Red G, [Strain = WT diploid], [Gamma exposure (krad) = 80], [Isolate type = Can1-Red]; Can1-White NG, [Strain = WT diploid], [Gamma exposure (krad) = 0], [Isolate type = Can1-White]; Can1-White G, [Strain = WT diploid], [Gamma exposure (krad) = 80], [Isolate type = Can1-White]; Small NG, [Strain = WT diploid], [Gamma exposure (krad) = 0], [Isolate type = small]; Small G, NR, [Strain = WT diploid], [Gamma exposure (krad) = 80], [Isolate type = small_not-rearranged]; Small G,R, [Strain = WT diploid], [Gamma exposure (krad) = 80], [Isolate type = small_rearranged]; Large NG, [Strain = WT diploid], [Gamma exposure (krad) = 0], [Isolate type = Large]; Large G, [Strain = WT diploid], [Gamma exposure (krad) = 80], [Isolate type = Large]. S2D Table, *P*-values for statistical evaluation of incidence of gamma-induced clusters in WT diploid yeast; S2E Table: *P*-values for statistical evaluation of incidence of CG single-switch clusters and C- or G-coordinated cluster in haploid yeast; S2F Table, cluster details for diploid Can1-Red yeast isolate 7_24_CA_2 shown in Fig 2B. Column title "Reference_Allele" refers to the original nucleotide that was mutated. Column title "Tumor_Seq_Allele2" refers to the nucleotide change of the mutation in the cluster. Description of all column titles can be found in S3 Table. Source data for Fig 2B. S2G Table, cluster IDs and cluster types of cluster examples shown in Fig 4A. The column titled "StrainCluster_ID" corresponds to the same column in S3 Table and in S1 Data. Source data for Fig 4A. S2H Table, cluster type distribution in different isolate types of diploid yeast and statistical comparisons between C- or G-coordinated versus CG single-switch cluster distributions. Source data for Fig 4B. S2I Table, source data for Fig 5B, 5C and 5D showing cluster type distribution in haploid yeast; S2J Table, cluster length and mutation density in individual haploid samples and statistical comparisons for C- or G-coordinated cluster length and density with Bonferroni correction for multiple hypothesis testing. Source data for S4A Fig. Values in "Cluster_Length" column for the scatterplot can be obtained by filtering in the column "Strain" by value corresponding to each group listed under x-axis. Source data for S4B Fig. Values in "Mutation_Density_per_kb" column for the scatterplot can be obtained by filtering in the column "Strain" by value corresponding to each group listed under x-axis. CG, C- and/or G-containing; G, gamma-exposed; NG, non-gamma–exposed; WT, wild type.
(XLSX)

**S3 Table. Clusters identified by whole-genome sequencing of yeast isolates.** Datasheets in S3 Table: S3A Table, complete list of clusters identified by whole-genome sequencing of yeast isolates. Source data for S3A Fig (yeast). Values in "Mutation_Density_per_kb" column for the scatterplot can be obtained by filtering in the following columns for each group listed under x-axis after applying the following filters: for diploid yeast, [Strain = WT diploid], [Gamma exposure (krad) = 80], [spans_reporter = no], [CG_cluster_classification >3 mutations = cluster type corresponding to the cluster type in the scatter plot]; for haploid yeast, [Strain = WT haploid], [Gamma exposure (krad) = 40], [spans_reporter = no], [CG_cluster_classification >3 mutations = cluster type corresponding to the cluster type in the scatterplot]. Source data for S3B Fig (yeast). Values in "Cluster_Size_Complexes", column for each group listed under x-axis of the scatterplot can be obtained by applying the same filtering as described above for source data for S3A Fig (yeast). Source data for S3C Fig (yeast). Values in "Cluster_Length" column for each group listed under x-axis of the scatterplot can be obtained by applying the same filtering as described above for source data for S3A Fig (yeast). S3B Table, separation of terminal mutations from the rest of a cluster in the cluster category II (coordinated with terminal CG). Source data for S3D Fig (yeast). Values for yeast scatterplots are provided in "Distance_terminal_non_matched" and in "Distance_terminal_matched" columns. CG, C- and/or G-containing; WT, wild type.
(XLSX)

**S4 Table. Details of hypermutable ssDNA per genome in yeast and in cancers and relation to genomic features.** Datasheets in S4 Table: S4A Table, sum cluster length and percent ssDNA using CG clusters >3 mutations in WT haploid and diploid Can1-Red colonies. Source data for yeast portions of Fig 3A and 3B. Values for haploid and for diploid yeast can be obtained from columns "Sum_Cluster_Length" and "percent_ssDNAperGenome" by filtering "Ploidy" column using "Haploid" or "Diploid" filters, respectively. S4B Table, individual yeast isolate details containing sum cluster length, total ssDNA per genome, sum cluster types, sum cluster size, and sum cluster density. S4C Table, sum cluster length, percent ssDNA for highly APOBEC-enriched cancer tumor samples with annotations for UNG1. Source data for cancer portion of Fig 3A and 3B. Values for cancer scatter plots can be obtained directly from columns "Sum_Cluster_Length" and "percent_ssDNAperDIPLOIDGenome" S4D Table, Pearson correlation of cluster features in diploid Can1-Red colonies; S4E Table, CNV outputs in haploid strains with details of overlapping clusters. S4F Table, distributions of nonselected clusters across chromosomes in WT diploid. Source data for S6 Fig. S4G Table, ratios of track lengths between the left side and right side of single-switch clusters in haploid yeast. APOBEC, apolipoprotein B mRNA editing enzyme, catalytic polypeptide-like; CG, C- and/or G-containing; CNV, copy number variation; ss, single-stranded; WT, wild type.
(XLSX)

**S5 Table. Clusters identified in highly APOBEC-enriched cancers.** Datasheets in S5 Table: S5A Table, complete list of clusters identified in highly APOBEC-enriched cancers. Source data for S3A Fig (cancer). Values in "Mutation_Density_per_kb" column for the scatterplot can be obtained for each group listed under x-axis by applying the following filter: [CG_cluster_classification >3 mutations = cluster type corresponding to the cluster type in the scatter plot]; source data for S3B Fig (cancer). Values in "Cluster_Size_Complexes" column for each group listed under x-axis of the scatterplot can be obtained by applying the same filtering as described above for source data for S3A Fig (cancer). Source data for S3C Fig (cancer). Values in "Cluster_Length" column for each group listed under x-axis of the scatterplot can be obtained by applying the same filtering as described above for source data for S3A Fig (cancer). S5B Table, separation of terminal mutations from the rest of a cluster in the cluster category II

(coordinated with terminal CG) cancer data. Source data for S3D Fig (cancer). Values for cancer scatterplots are provided in "Distance_terminal_non_matched" and in "Distance_terminal_matched" columns. APOBEC, apolipoprotein B mRNA editing enzyme, catalytic polypeptide-like; CG, C- and/or G-containing.
(XLSX)

**S6 Table. Mutagenesis in highly APOBEC-enriched cancers.** Datasheets in S6 Table: S6A Table, APOBEC enrichment values for "sum_cluster" = APOBEC enrichment values in clustered mutations only; "sum_scattered" = APOBEC enrichment values in scattered mutations only; "sum_all" = APOBEC enrichment values in all genome-wide mutations (scattered and clustered combined). S6B Table, the number of CG clusters with >3 mutations in a cluster per tumor sample. Note column entitled "number_CandOrG_containing_Clusters_gr3MutClus" refers to the number of CG clusters with >3 mutations. Source data for Fig 6A. Values for each cancer type on the scatterplot can be obtained by filtering applied to "Cancer_type" column. S6C Table, the distribution of the different types of CG clusters in highly APOBEC-enriched cancer types. Source data for Fig 6B, 6C and 6D. Fractions (percent) shown in Fig 6B were calculated taking the value in "Sum clusters" column as 100%. Fractions (percent) shown in Fig 6C were calculated taking the total value of "5′ C Single-switch clusters" and "5′ G Single-switch clusters" columns as 100%. Fractions (percent) shown in Fig 6D were calculated taking the total value of "C-coordinated adjacent to a single G" and "G-coordinated adjacent to a single C" columns as 100%. All values taken as 100% also shown on top of corresponding bars in Fig 6. APOBEC, apolipoprotein B mRNA editing enzyme, catalytic polypeptide-like; CG, C- and/or G-containing.
(XLSX)

**S7 Table. Primers for strain construction and verification.**
(XLSX)

**S8 Table. Yeast strains.**
(XLSX)

**S1 Data. Files listing mutation calls and cluster calls in sequenced yeast genomes and detailed summaries of APOBEC signature analysis in whole genome and cluster calls in human cancers.** APOBEC, apolipoprotein B mRNA editing enzyme, catalytic polypeptide-like.
(ZIP)

## Acknowledgments

The results published here are partly based upon data generated by The Cancer Genome Atlas and obtained from the Database of Genotypes and Phenotypes (dbGaP) with accession number phs000178.v8.p7. We thank Ms. Morgan Landis and Mr. Alan Zhao for help with experiments, Mr. Gregory Stamper for help in setting data networking, and Drs. Natalya Degtyareva and Scott Lujan for advice on the manuscript.

## Author Contributions

**Conceptualization:** Cynthia J. Sakofsky, Dmitry A. Gordenin.

**Data curation:** Cynthia J. Sakofsky, Dmitry A. Gordenin.

**Formal analysis:** Cynthia J. Sakofsky, Natalie Saini, Dmitry A. Gordenin.

**Investigation:** Cynthia J. Sakofsky, Kin Chan, Ewa P. Malc, Piotr A. Mieczkowski.

**Methodology:** Cynthia J. Sakofsky, Kin Chan, Dmitry A. Gordenin.

**Software:** Leszek J. Klimczak, Adam B. Burkholder, David Fargo.

**Supervision:** Dmitry A. Gordenin.

**Writing – original draft:** Cynthia J. Sakofsky, Dmitry A. Gordenin.

**Writing – review & editing:** Cynthia J. Sakofsky, Natalie Saini, Dmitry A. Gordenin.

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
