## [Editor Report · Decision Letter 0]

24 Jul 2019

Dear Dr Gordenin, 

Thank you for submitting your manuscript entitled "Repair of multiple simultaneous double-strand breaks causes bursts of genome-wide clustered hypermutation" for consideration as a Research Article by PLOS Biology.

Your manuscript has now been evaluated by the PLOS Biology editorial staff as well as by an academic editor with relevant expertise and I am writing to let you know that we would like to send your submission out for external peer review.

*Please be aware that, due to the voluntary nature of our reviewers and academic editors, manuscripts may be subject to delays during this busy summer travel season. Thank you for your patience.*

**Important**: Please also see below for further information regarding completing the MDAR reporting checklist. The checklist can be accessed here: https://plos.io/MDARChecklist

Please re-submit your manuscript and the checklist, within two working days, i.e. by Jul 26 2019 11:59PM.

Kind regards,

Hashi Wijayatilake, PhD,

Managing Editor

PLOS Biology

INFORMATION REGARDING THE REPORTING CHECKLIST:

PLOS Biology is pleased to support the "minimum reporting standards in the life sciences" initiative (https://osf.io/preprints/metaarxiv/9sm4x/). This effort brings together a number of leading journals and reproducibility experts to develop minimum expectations for reporting information about Materials (including data and code), Design, Analysis and Reporting (MDAR) in published papers. We believe broad alignment on these standards will be to the benefit of authors, reviewers, journals and the wider research community and will help drive better practise in publishing reproducible research. 

We are therefore participating in a community pilot involving a small number of life science journals to test the MDAR checklist. The checklist is intended to help authors, reviewers and editors adopt and implement the minimum reporting framework. 

IMPORTANT: We have chosen your manuscript to participate in this trial. The relevant documents can be located here:

MDAR reporting checklist (to be filled in by you): https://plos.io/MDARChecklist

**We strongly encourage you to complete the MDAR reporting checklist and return it to us with your full submission, as described above. We would also be very grateful if you could complete this author survey:

https://forms.gle/seEgCrDtM6GLKFGQA

Additional background information:

Interpreting the MDAR Framework: https://plos.io/MDARFramework

Please note that your completed checklist and survey will be shared with the minimum reporting standards working group. However, the working group will not be provided with access to the manuscript or any other confidential information including author identities, manuscript titles or abstracts. Feedback from this process will be used to consider next steps, which might include revisions to the content of the checklist. Data and materials from this initial trial will be publicly shared in September 2019. Data will only be provided in aggregate form and will not be parsed by individual article or by journal, so as to respect the confidentiality of responses. 

Please treat the checklist and elaboration as confidential as public release is planned for September 2019.

We would be grateful for any feedback you may have.

---

## [Decision Letter · Decision Letter 1]

13 Aug 2019

Dear Dr Gordenin,

Thank you very much for submitting your manuscript "Repair of multiple simultaneous double-strand breaks causes bursts of genome-wide clustered hypermutation" for consideration as a Research Article by PLOS Biology. As with all papers reviewed by the journal, yours was evaluated by the PLOS Biology editors as well as by an Academic Editor with relevant expertise and by independent reviewers. As you can see from the reviews appended below, the reviewers all appreciated the attention to this topic and are generally very supportive of the manuscript. Based on the reviews, we will probably accept this manuscript for publication, providing that you will revise and clarify the manuscript according to the review recommendations. Please also see below for Data Policy-related requests.

We expect to receive your revised manuscript within two weeks. Your revisions should address the specific points made by each reviewer. In addition to the remaining revisions and before we will be able to formally accept your manuscript and consider it "in press", we also need to ensure that your article conforms to our guidelines. A member of our team will be in touch shortly with a set of requests. As we can't proceed until these requirements are met, your swift response will help prevent delays to publication.

Please note that you may have the opportunity to make the peer review history publicly available. The record will include editor decision letters (with reviews) and your responses to reviewer comments. If eligible, we will contact you to opt in or out.

Early Version

Sincerely,

Hashi Wijayatilake, PhD, 

Managing Editor

PLOS Biology

You are aware of the PLOS Data Policy, which requires that all data be made available without restriction: http://journals.plos.org/plosbiology/s/data-availability. For more information, please also see this editorial: http://dx.doi.org/10.1371/journal.pbio.1001797

Althoug we note that you might have included the data requested in the supplementary files and tables, we do require all individual quantitative observations that underlie the data summarized in the figures and results of your paper be made available in one of the following forms:

**Thank you for providing the supplemental data file. Please double check and ensure that you provide the individual numerical values that underlie the summary data displayed in the following figure panels. Please also label the files/sheets/tabs clearly indicating where the data for each following figure panel can be found:

Fig. 2A, C, D; Fig. 3A, B; Fig. 4B; Fig. 5A, B, C, D; Fig. 6A, B, C, D; Fig. S2A, B; Fig. S3A, B, C; Fig. S4A, B and Fig. S6

** Please also ensure that figure legends in your manuscript include information on were the underlying data can be found.

Please ensure that your Data Statement in the submission system accurately describes where your data can be found (including mention of the supplemental data file).

For manuscripts submitted on or after 1st July 2019, we require the original, uncropped and minimally adjusted images supporting all blot and gel results reported in an article's figures or Supporting Information files. We will require these files before a manuscript can be accepted so please prepare them now, if you have not already uploaded them. Please carefully read our guidelines for how to prepare and upload this data: https://journals.plos.org/plosbiology/s/figures#loc-blot-and-gel-reporting-requirements.

REVIEWS:

Reviewer 1:

Cancer genomes can exhibit clusters of mutations that are proposed to be caused by the APOBEC3s, a family of cytidine deaminases that preferentially target single-stranded DNA (ssDNA). This manuscript by Sakofsky et al seeks to identify the causes of ssDNA accumulation - and potential APOBEC mutagenesis - in cancer genomes. This is an important question as previous work from this group shows that persistent ssDNA - rather than the levels of A3s - can be the limiting factor for mutation cluster formation (Chan et al, 2015). The authors propose that DNA double strand breaks (DSBs) enable cluster formation through the generation of ssDNA during the repair process. The authors tackle this problem using a clever triple reporter system to identify yeast strains most likely to exhibit mutation clusters (selected mutation clusters are predictive of the presence of non-selected mutation clusters elsewhere in the genome). In their setup, DSBs are caused by gamma-irradiation. Using a series of yeast strains defective in specific repair pathways, the authors identify 3 pathways that contribute to cluster formation at DSBs: 5’-3’ bi-directional resection, uni-directional resection, and break-induced replication (BIR). This is a substantive finding because it supports the idea that the availability of ssDNA is an important determinant in directing APOBEC mutagenesis. Finally, the authors undertake comparative analyses between APOBEC hyper-mutated cancer genomes and model yeast strains and conclude that cluster formation in many cancers is driven by a BIR-like mechanism. 

The authors proposal that clustered mutagenesis in human cancer is driven by a BIR-like mechanism is interesting, but requires further investigation. The mechanics of BIR in mammalian systems have not been well defined and may not be accurately defined by the yeast data. Throughout most of the text the authors are appropriately careful in their interpretation. Given the limited support for a BIR-like mechanism in human cancer, the authors discuss potential alternate sources of ssDNA in cancer genomes, including long-range resection and telomere protection crisis. 

This is an interesting study that makes important observations regarding the genesis of APOBEC mutations in cancer genomes. I have only a few comments and questions: 

Major points 

The authors find that there is no strong bias for 5’C to 3’G versus 5’G to 3’C in single switch clusters in cancer genomes. This leads them to propose that single-switch clusters in cancer are not formed by bi-directional resection. CG single-switch clusters can arise following bidirectional resection and correct repair of a DNA DSB. Is it possible to differentiate between a BIR-like mechanism and an aborted repair mechanism that was initiated by bi-directional resection, but associated with loss of copy number at one side of the DNA DSB? 

The authors use mutation cluster lengths to infer regions of ssDNA in mutated yeast strains and APOBEC-enriched human cancer types (Figure 3). As the authors admit, their analysis likely underestimates the extent of ssDNA formation. Unless the authors more directly measure ssDNA formation they should substitute a title that more accurately reflects what they are measuring. 

The authors exclusively focus on APOBEC3A in their analysis, but APOBEC3B has also been implicated in cancer-associated mutagenesis. Can APOBEC3B also generate mutation clusters in their system? 

Minor points 

Were CG multiple-switch clusters included in the ssDNA analysis (Figure 3)? Inclusion of these clusters may inflate ssDNA estimates due to the acknowledged possibility that these clusters may reflect clusters that arose independently on different copies of the homologous chromosomes. 

“Haploid” is mis-spelled in Fig. 5A x axis.

---

Reviewer 2:

The authors examine mutations formed after gamma irradiation using selected and unselected segregants in diploid yeast. They look at mutations in yeast strains expressing APOBEC3A and observe clustered mutations similar to those seen in tumor cells. They suggest that the mutations arise from repair pathways that generate long stretches of single-strand DNA from processing of double-strand breaks. For proof of this they examine mutagenesis in yeast mutants that impair production of single-strand DNA in various DNA double-strand break repair pathways. From this type of analysis the authors conclude that clustered mutagenesis arises from a BIR type of mechanism. 

The work is very carefully done and the manuscript contains a tremendous amount of information, especially in the supplemental tables. Although other groups have published that DNA double-strand break repair in yeast is mutagenesis and the Gordenin groups have previously shown that there are clustered mutations in yeast, with a role for BIR in their generation, this paper draws together both BIR and APOBEC3A in clustered mutagenesis in yeast. It would be important to emphasize first, what is new in this paper and second, as the authors propose that this is a model for clustered mutation in cancer cells from DSB repair, when during the growth of a tumor cell there would be bursts of DSBs associated with long ssDNA repair intermediates. At the end the authors suggest that a single DSB could be the initiating event but do not address this experimentally. 

On page 5 the authors discuss DSB repair models that form long ssDNA intermediates and refer to Figure 1. In the text they discuss BIR after bi-directional resection but show BIR first in the Figure. The text and figure could be better coordinated. The authors could also mention in what context there could be uni-directional resection, especially as the DSBs induced in this study for determination of mutations is by gamma irradiation. Would this be inducing both DSB repaired by unidirectional resection and bi-directional resection but only the uni-directional resection events in the presence of APOBEC3A give the clustered mutations? That should be made clear. 

Page 6, middle. Here the authors mention bursts of DSBs in cancer genomes. How do these arise? Spontaneously, replication stress, radiation treatment? 

Figure 2 and text. Although the authors denote the repair segregants as Can1-Red, it would be helpful to overtly state that the strain is initially sensitive to canavanine and white. The authors refer to a three-gene reporter at times, but never mention the URA3 gene or its utility in the reporter. Is it necessary to show it? 

APOBEC3A cytosines. On page 1 the authors say that the characteristic trinucleotide context for mutagenesis is tCw. On page 8 they say it is yCa. Please be consistent. As an aside, from the sequencing were the CAN1 and ADE2 inactivating mutations determined by sequencing an A3A signature? 

Figure 4 and page 12. On page 12 three types of mutation clusters are described (I-III) but Figure 4A shows 4 types labeled as I-IV. This is really confusing. All the graphs show 4 types of clusters. 

Figure 5. “haploid’ is misspelled. It is difficult to see the “difference” om pol32 vs sgs1 pol32 frequencies as the Y-axis break seems to be in that value. Could the fold difference from WT be indicated above each plot? 

The discussion needs to refer back to the ssDNA generating models shown in Figure 1. 

There are some added or missing articles (an, the, etc) in the manuscript. 

Page 24 last line, “mutagenesis” is misspelled.

---

Reviewer 3:

This is a nicely presented and expertly performed study, where yeast experiments give some further mechanistic clues regarding APOBEC mutation clusters, which are already a well-established feature of cancer genomes. Specifically, APOBEC clustered mutations in tumors are here shown to have characteristics similar to BIR-related APOBEC clusters artificially induced in yeast. Although the same speculation can possibly be made based on the appearance of human APOBEC clusters even without the yeast experiments presented here, I do believe the results add to the understanding of APOBEC mutagenesis. It can be noted that the system used here is unnatural in the sense that is based on a UNG1 -/- background, thus altering and amplifying the impact of APOBEC on ssDNA. All together, the manuscript is of high quality and gives the impression of already having been through revision. 

All raw data appears available and deposited in public repositories and statistics appears adequate. 

Minor points: 

Regarding clusters harboring single non-coordinated mutations, as noted this could be either related to the underlying repair event or alternatively simply random colocalization. This could perhaps be tested statistically, i.e. given the overall burden and signature, is it unexpected to observe these single mutations overlapping with the clusters to the extent seen here? 

CG clusters were somewhat arbitrarily required to have >3 mutations. Is this always stringent enough (e.g. in a high-burden tumor genome) to ensure non-random cooccurrence representing true clustering? The human genome is large, presenting lots of opportunities for random colocalization. 

Page 6 - should say “sufficiently persistently”? 

Page 4, mutahgenesis

---

## [Editor Report · Decision Letter 2]

12 Sep 2019

Dear Dr Gordenin,

On behalf of my colleagues and the Academic Editor, Scott Keeney, I am pleased to inform you that we will be delighted to publish your Research Article in PLOS Biology. 

Early Version

PRESS 

Kind regards,

Sofia Vickers

Senior Publications Assistant

PLOS Biology

On behalf of, 

Hashi Wijayatilake,

Managing Editor

PLOS Biology